# Geometry-Aware Adaptation for Pretrained Models

**Nicholas Roberts, Xintong Li, Dyah Adila, Sonia Cromp,**
**Tzu-Heng Huang, Jitian Zhao, Frederic Sala**
University of Wisconsin-Madison
{nick11roberts, fredsala}@cs.wisc.edu
{xli2224, adila, cromp, thuang273, jzhao326}@wisc.edu

## Abstract

Machine learning models—including prominent zero-shot models—are often trained on datasets whose labels are only a small proportion of a larger label space. Such spaces are commonly equipped with a metric that relates the labels via distances between them. We propose a simple approach to exploit this information to adapt the trained model to reliably predict new classes—or, in the case of zero-shot prediction, to improve its performance—without any additional training. Our technique is a drop-in replacement of the standard prediction rule, swapping $\arg\max$ with the Fréchet mean. We provide a comprehensive theoretical analysis for this approach, studying (i) learning-theoretic results trading off label space diameter, sample complexity, and model dimension, (ii) characterizations of the full range of scenarios in which it is possible to predict any unobserved class, and (iii) an optimal active learning-like next class selection procedure to obtain optimal training classes for when it is not possible to predict the entire range of unobserved classes. Empirically, using easily-available external metrics, our proposed approach, LOKI, gains up to 29.7% relative improvement over SimCLR on ImageNet and scales to hundreds of thousands of classes. When no such metric is available, LOKI can use self-derived metrics from class embeddings and obtains a 10.5% improvement on pretrained zero-shot models such as CLIP.

## 1   Introduction

The use of pretrained models is perhaps the most significant recent shift in machine learning. Such models can be used out-of-the-box, either to predict classes observed during pretraining (as in ImageNet-trained ResNets [14]) or to perform zero-shot classification on any set of classes [32]. While this is exciting, label spaces are often so huge that models are unlikely to have seen *even a single point* with a particular label. Without additional modification, pretrained models will simply fail to reliably predict such classes. Instead, users turn to fine-tuning—which requires additional labeled data and training cycles and so sacrifices much of the promise of zero-shot usage.

How can we adapt pretrained models to predict new classes, without fine-tuning or retraining? At first glance, this is challenging: predictive signal comes from labeled training data. However, *relational* information between the classes can be exploited to enable predicting a class even when there are no examples with this label in the training set. Such relational data is commonly available, or can be constructed with the help of knowledge bases, ontologies, or powerful foundation models [39].

How to best exploit relational structure remains unclear, with a number of key challenges: We might wish to know what particular subset of classes is rich enough to enable predicting many (or all) remaining labels. This is crucial in determining whether a training set is usable or, even with the aid of structure, insufficient. It is also unclear how approaches that use relational information interact with the statistical properties of learning, such as training sample complexity. Finally, performing adaptation requires an efficient and scalable algorithm.

37th Conference on Neural Information Processing Systems (NeurIPS 2023).

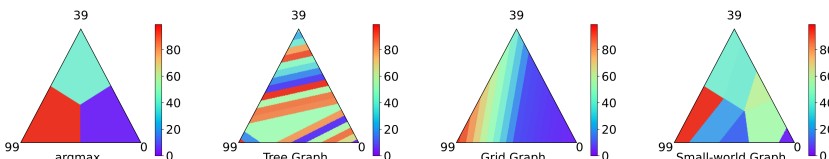

Figure 1: Classification regions in the probability simplex of 3-class classifiers faced with a 100-class problem. The probability simplex using $\arg\max$ prediction can only output one of three classes. LOKI uses the entire probability vector to navigate the class metric space, leading to more prediction regions. (Left) regions from $\arg\max$ prediction. (Centers, Right) classification regions from LOKI.

This work addresses these challenges. It proposes a simple and practical approach to learning in structured label spaces, with theoretical guarantees. First, we offer a simple way to translate the soft outputs (i.e., probability vectors) produced by *any* supervised learning model into a more general model that can exploit geometric information for label structure. In other words, our approach, called LOKI,[1] is a simple *adaptor* for pretrained models. LOKI can be applied via a fixed linear transformation of the model outputs. LOKI's simplicity makes it applicable to a broad range of settings while enabling very high-cardinality predictions subject to a potentially small model output budget—we provide a visualization of this key idea in Figure 1.

Theoretically, we provide a rich set of results for the metric-based adaptation setting. First, we introduce a learning-theoretic result in terms of the sample complexity of the pretrained model. It captures a key tradeoff between the number of classes and metric structure, the problem dimensionality, and the number of samples used to train the model prior to adaptation. Next we exhaustively study the properties of training sets, determining for a wide class of relational structures the minimum number (and structure) of subsets that enable reliable prediction. Finally we show how to exploit this result in an active learning-like approach to selecting points that will improve deficient training datasets.

Experimentally, we show that using structural information improves prediction in high-cardinality settings. We demonstrate the strength of the active learning-based approach to dataset expansion over random baselines. Finally, and most excitingly, we show that even in zero-shot models like CLIP, where it is possible to produce probability vectors over any possible class, the use of our adaptor leads to a **19.53%** relative improvement.

## 2   Background

First, we introduce the problem setting, notation, and mathematical tools that we will use. Afterward, we discuss how LOKI relates to prior work.

**Problem Setting**   As in conventional supervised learning, we have a dataset $(x_1, y_1), \ldots, (x_n, y_n)$ drawn from a distribution on $\mathcal{X} \times \mathcal{Y}$, where $\mathcal{X}$ and $\mathcal{Y}$ are the input and label spaces. In our setting, $N := |\mathcal{Y}|$ is finite but large; often $N \gg n$—so that many labels will simply not be found in the training dataset. We let $\Lambda \subseteq \mathcal{Y}$ with $|\Lambda| = K$ be the set of observed labels. For convenience of notation, we also define $\boldsymbol{\lambda} := [\lambda_i]_{i=1}^K$, $\boldsymbol{y} := [y_i]_{j=1}^N$ to be the vectors of elements in $\Lambda$ and $\mathcal{Y}$.

In addition to our dataset, we have access to a relational structure on $\mathcal{Y}$. We assume that $\mathcal{Y}$ is a metric space with metric (distance) $d : \mathcal{Y}^2 \to \mathbb{R}$; $d$ encodes the relational structure of the label space. Specifically, we model this metric space using a graph, $G = (\mathcal{Y}, \mathcal{E})$ where $\mathcal{E} \subseteq \mathcal{Y}^2 \times \mathbb{R}_+$ is a set of edges relating the labels and the standard shortest-path distance $d : \mathcal{Y}^2 \to \mathbb{R}_{\geq 0}$. In addition to its use in prediction, the metric $d$ can be used for evaluating a model by measuring, for example, $\frac{1}{n} \sum_{i=1}^n d^2(f(x_i), y)$—the analogue of the empirical square loss.

**Fréchet Mean Estimator**   Drawing on ideas from structured prediction [5, 35], we use a simple predictor that exploits the label metric. It relies on computing the *Fréchet mean* [8], given by

$$m_{\boldsymbol{y}}(\boldsymbol{w}) := \arg\min_{y \in \mathcal{Y}} \sum_{i=1}^K \boldsymbol{w}_i d^2(y, \boldsymbol{y}_i), \tag{1}$$

---

[1]Refers to the 'locus' (plural: *loci*) of the Fréchet mean.

where $\boldsymbol{w} \in \mathbb{R}_{\geq 0}^K$ is a set of weights. The Fréchet mean generalizes the barycenter to metric spaces and is often used in geometric machine learning [26].

**Locus of the Fréchet mean.** The *locus of the Fréchet mean* is the set of all Fréchet means under different weights [29]. We write it as $\Pi(\boldsymbol{y}) := \cup_{\boldsymbol{w} \in \Delta^{K-1}} m_{\boldsymbol{y}}(\boldsymbol{w})$.

$\Pi(\boldsymbol{y})$ can be thought of the set of all labels in $\mathcal{Y}$ that are reachable by the Fréchet mean given $\{\boldsymbol{y}_i\}_{i=1}^K \subseteq \mathcal{Y}$ and different choices of its parameter $\boldsymbol{w}$. Intuitively, we can think of the locus for a given dataset as describing how usable it is for predicting beyond the observed labels. Trivially, if $\{\boldsymbol{y}_i\}_{i=1}^K = \mathcal{Y}$, then $\Pi(\boldsymbol{y}) = \mathcal{Y}$. We are primarily interested in the case in which $\{\boldsymbol{y}_i\}_{i=1}^K \subset \mathcal{Y}$ yet we still have $\Pi(\boldsymbol{y}) = \mathcal{Y}$, or that $|\Pi(\boldsymbol{y})|$ is at least sufficiently large.

## 2.1 Relation to Prior work

LOKI is primarily related to two areas: zero-shot learning and structured prediction.

**Zero-Shot Learning** Like zero-shot learning (ZSL), LOKI is capable of predicting unobserved classes. Our framework is most closely related to *generalized* ZSL, which uses side information to predict both observed and unobserved classes. Many types of external knowledge are used in ZSL, including text [28, 38, 9], attributes [22, 23, 7], knowledge graphs [42, 34, 10], ontologies [25], and logical rules [33]. Our work is most closely related to ZSL approaches that rely on knowledge graph information. Often, ZSL methods that use knowledge graph information rely on the use of graph neural network architectures [42, 17, 18]. *However, we note that these architectures can be heavyweight and can be challenging to scale up to extremely large graphs, whereas* LOKI *does not have architectural requirements and scales linearly in the size of the graph when $K$ is small.*

**Structured Prediction** Structured prediction (SP) operates in label spaces that are endowed with algebraic or geometric structure [1, 21]. This includes problems such as predicting permutations [19], non-Euclidean and manifold regression [31, 36], and learning on graphs [12]. LOKI is the most immediately related to the latter, however, any finite metric space can be represented as a graph, which further lends to the flexibility of our approach. Even in discrete structured prediction settings, the cardinality of the label space may be combinatorially large. *As such,* LOKI *can be viewed as a simple method for adapting classifiers to structured prediction.*

The Fréchet mean has been used in structured prediction—but in approaches requiring training. In [5], $\hat{f}(x) = \arg\min_{y \in \mathcal{Y}} \frac{1}{n} \sum_{i=1}^n \alpha_i(x) d^2(y, \boldsymbol{y}_i)$, where $\alpha(x) = (K + n\nu I)^{-1} K_x$. $K$ is the kernel matrix for a kernel $k : \mathcal{X} \times \mathcal{X} \to \mathbb{R}$, so that $K_{i,j} = k(x_i, x_j)$, $(K_x)_i = k(x, x_i)$. $\nu$ is a regularization parameter. In other words, the weight $w_i$ corresponding to $\boldsymbol{y}_i$ is the average produced by solving a kernel regression problem at all points $x_k$ in the dataset where $y_k = \boldsymbol{y}_i$. It has also used in weak supervision (WS) for metric-equipped label spaces [41, 37], where the goal is to produce labeled data for training structured prediction models.

## 3 Framework

We introduce our framework—LOKI. We show how LOKI can be used to adapt any supervised classifier over a set of $K$ classes to a much richer set of possible class predictions. It does so by weighting the Fréchet mean by the classifier's per-class prediction probabilities or logits, allowing it to predict any class in the locus of the Fréchet mean—potentially far more classes than the initial $K$. Next, we show how LOKI can be expressed as a *fixed* linear transformation of a model's outputs. Finally, we show that LOKI relates to standard classification.

### 3.1 LOKI: Adapting Pretrained Models

We describe our approach to adapting pretrained classifiers–trained on a set of classes $\Lambda$—to the metric geometry of the label space $\mathcal{Y}$, enabling the prediction of unobserved classes.

We model unobserved classes $\mathcal{Y} \setminus \Lambda$ using the Fréchet mean among observed classes weighted by their *prediction probabilities* $P(y = \lambda_i | x$ and $y \in \Lambda)$. We denote the vector of model outputs as

$\mathbf{P}_{y|x} := [\mathbf{P}_{\lambda_i|x}]_{i=1}^K = [P(y = \lambda_i|x \text{ and } y \in \Lambda)]_{i=1}^K$. Then predictions using LOKI are given by

$$\hat{y} \in m_{\boldsymbol{\lambda}}(\mathbf{P}_{y|x}) = \arg\min_{y \in \mathcal{Y}} \sum_{i=1}^K \mathbf{P}_{\lambda_i|x} d^2(y, \lambda_i).$$

## 3.2  LOKI as a linear transformation of model outputs

Most standard classifiers output a vector of prediction probabilities, $\mathbf{P}_{y|x}$, whose entries correspond to the confidence of predicting a specific class. Predictions are typically given by $\hat{y} \in \arg\max_{i \in [K]}(\mathbf{P}_{y|x})_i$. LOKI generalizes this prediction rule when viewed as a linear transformation of $\mathbf{P}_{y|x}$. Consider the LOKI prediction rule $\hat{y} \in m_{\boldsymbol{\lambda}}(\mathbf{P}_{y|x}) = \arg\min_{y \in \mathcal{Y}} \sum_{i=1}^K \mathbf{P}_{\lambda_i|x} d^2(y, \lambda_i) = \arg\max_{j \in [N]}(\mathbf{D}\mathbf{P}_{y|x})_j$, where $\mathbf{D}_{j,i} = [-d^2(y_j, \lambda_i)]$; $\mathbf{D} \in \mathbb{R}^{N \times K}$ is the matrix of negative squared distances between the observed classes and the rest of the label space. Thus LOKI can be used within standard classification pipelines when the model output $\mathbf{P}_{y|x}$ is multiplied by the fixed matrix $\mathbf{D}$.

## 3.3  Generalizing Standard Classification

We provide a simple intuition for our approach. The fact that LOKI reduces to standard classification among the observed classes has several implications. This includes the idea that under our framework, forms of few-shot, zero-shot, hierarchical, and partial label learning all reduce to standard classification when additional metric information is introduced.

**Generalizing the arg max prediction rule**  In the absence of this metric information—a situation that we model using the complete graph and setting $\Lambda = \mathcal{Y}$—our framework also recovers standard classification. Indeed, both in terms of error modeling and in terms of inter-class similarity, the intuition of standard classification and of the 0-1 loss are captured well by the unweighted complete graph—simply treat all differences equally. This graph is given as $K_N := (\mathcal{Y}, \mathcal{Y}^2 \times \{1\})$—i.e., every label is equidistant from every other label. Plugging this into LOKI, we obtain the following:

$$\hat{y} \in m_{\boldsymbol{\lambda}}(\mathbf{P}_{y|x}) = \arg\min_{y \in \mathcal{Y}} \sum_{i=1}^K \mathbf{P}_{\lambda_i|x} d^2(y, \lambda_i) = \arg\min_{y \in \mathcal{Y}} \sum_{i=1}^K \mathbf{P}_{\lambda_i|x} \mathbf{1}\{y \neq \lambda_i\} = \arg\max_{i \in [K]} \mathbf{P}_{\lambda_i|x},$$

which is exactly the standard classification prediction rule.

**Generalizing the 0-1 loss via the expected squared distance**  *The expected squared distance $\mathbb{E}[d^2(y, \hat{y})]$ is the standard loss function for many structured prediction problems.* Note that **accuracy fails in such settings**—since it cannot distinguish between small and large errors. This is most clearly seen in the extreme case of regression, where test accuracy will virtually always be zero no matter how good a trained model is. At the other extreme, the complete graph, this loss function becomes the standard 0-1 loss: $\mathbb{E}[d^2(y, \hat{y})] = \mathbb{E}[\mathbf{1}\{y \neq \hat{y}\}]$. As an adaptor for structured label spaces, we use the empirical version of this loss to evaluate LOKI. Note that the expected squared distance subsumes other metrics as well. For example, when $\mathcal{Y} = \mathbb{R}$, we can derive the standard MSE by setting $d(y, \hat{y}) = |y - \hat{y}|$, which is just the standard L1 distance metric. Other scores such as recall at top-k can be similarly obtained at the cost of $d(\cdot, \cdot)$ being a true metric. In other words, $\mathbb{E}[d^2(y, \hat{y})]$ is an very general metric that supports any metric space, and we use it throughout this work.

## 4  Theoretical Results

**Challenges and Opportunities**  The $\arg\max$ of per-class model probabilities is a ubiquitous component of classification pipelines in machine learning. In order to predict unobserved classes using metric space information, LOKI replaces this standard component. As a simple but significant change to standard pipelines, LOKI opens up a new area for fundamental questions. There are three main flavors of theoretical questions that arise in this setting:

1. How does the performance of LOKI change as a function of the number of samples?
2. What minimal sets of observed classes are required to predict any class in the metric space?
3. How can we acquire new classes that maximize the total number of classes we can predict?

Excitingly, we provide a comprehensive answer to these questions for the most common metric spaces used in practice. First, we provide a general error bound in terms of the number of samples, observed classes, problem dimension, and the diameter of the metric space, that holds for any finite metric space. Second, we characterize the sets of observed classes that are required to enable prediction of any class, and show how this set differs for various types of metric spaces of interest. Finally, we provide an active learning algorithm for selecting additional classes to observe so as to maximize the number of classes that can be predicted and we characterize the types of metric spaces for which the locus can be computed efficiently. These results provide a strong theoretical grounding for LOKI.

## 4.1 Sample Complexity

What is the interplay between predicting unobserved classes based on metric space information and standard learning-theoretic notions like the sample complexity needed to train a model? Our first result illustrates this tradeoff, and relates it to the squared distance loss. Suppose that we only observe $K \leq N$ of the classes at training time, and that we fit a $K$-class Gaussian mixture model. We use LOKI to adapt our pretrained classifier to enable classification of any of the $N$ classes. We have that (a formal statement and interpretation are in the Appendix),

**Theorem 4.1** (**Informal** LOKI sample complexity). *Let $\mathcal{Y}$ be a set of classes represented by $d$ dimensional vectors under the Euclidean distance, and let $\Lambda \subseteq \mathcal{Y}$ be the set of $K$ observed classes. Assume that $n$ training examples are generated by an identity covariance Gaussian mixture model over classes $\Lambda$, and that test examples are generated over all classes $\mathcal{Y}$. Assume that we estimate a Gaussian mixture model on the training set and obtain probability estimates $\hat{\mathbb{P}}(y_i|x)$ for $i \in [K]$ for a sample $(x, y_*)$ from the test distribution. Then with high probability, under the following model,*

$$\hat{y}_* \in m_\Lambda([\hat{\mathbb{P}}(y_i|x)]_{i \in [K]}) = \arg\min_{y \in \mathcal{Y}} \sum_{i \in [K]} \hat{\mathbb{P}}(y_i|x) d^2(y, y_i)$$

*the sample complexity of estimating target $y_*$ from the test distribution $\mathcal{D}_{test}$ with prediction $\hat{y}_*$ is:*

$$\mathbb{E}_{(x, y_*) \sim \mathcal{D}_{test}}[d^2(y_*, \hat{y}_*)] \leq O\left(\frac{d}{\alpha}\sqrt{\frac{\log K/\delta}{n}}\left(\frac{1}{\left(R^{1-\frac{2}{d}} - \frac{\log R}{R}\right)} + \sqrt{d}\right)\right)$$

*where $\alpha$ is related to the sensitivity of the Fréchet variance to different node choices.*

## 4.2 Optimal Label Subspaces

Next we characterize the subset of distinct labels required to predict any label using our label model with respect to various types of metric spaces.

We first consider label spaces whose metric is a tree graph—such metrics are, for example, related to performing hierarchical classification and weak supervision (where only partial labels are available). We consider a special type of tree called a phylogenetic tree, in which only the leaves can be designated as labels—phylogenetic trees are commonly used to relate the labels of image classification datasets. Afterwards we perform a similar analysis for grid graphs, which are important for label spaces that encode spatial information. Finally, we discuss the case in which no useful metric information is available, i.e., the complete graph.

Our goal in this section is to characterize the properties and size of $\{\lambda_i\}_{i=1}^K$ in each of these metric spaces such that we still have $\Pi(\Lambda) = \mathcal{Y}$. We characterize 'optimal' subsets of classes in each of the spaces under a certain notions of optimality. We provide several relevant definitions pertaining to this concept, starting with a notion of being able to predict any possible class using observed classes.

**Definition 4.2** (Locus cover). Given a set $\Lambda \subseteq \mathcal{Y}$ for which we construct a tuple of its elements $\Lambda$, if it holds that $\Pi(\Lambda) = \mathcal{Y}$, then $\Lambda$ is a locus cover.

Definition 4.2 captures the main idea of LOKI—using some set of observed classes for which we can train classifiers, we would like to be able to predict additional unobserved classes using the geometry that relates the observed and unobserved classes. Namely, elements of $\Pi(\Lambda)$ are 'reachable' using LOKI. We refine this Definition to describe the trivial case that defaults to standard classification and the nontrivial case for which LOKI moves beyond standard classification.

**Definition 4.3** (Trivial locus cover). If $\Lambda = \mathcal{Y}$, then $\Lambda$ is the trivial locus cover.

This Definition captures the notion of observing all of the classes in the label space. Here, all of the elements of $\mathcal{Y}$ are trivially reachable using LOKI.

**Definition 4.4** (Nontrivial locus cover). A locus cover $\Lambda$ is nontrivial if $\Lambda \neq \mathcal{Y}$.

LOKI is more useful and interesting when faced with a nontrivial locus cover—under Definition 4.4, we can use some subset of classes $\Lambda$ to predict any label in $\mathcal{Y}$.

**Definition 4.5** (Minimum locus cover). Given a set $\Lambda \subseteq \mathcal{Y}$, if $\Lambda$ is the smallest set that is still a locus cover, then it is a minimum locus cover.

In cases involving an extremely large number of classes, it is desirable to use LOKI on the smallest possible set of observed classes $\Lambda$ such that all labels in $\mathcal{Y}$ can still be predicted. Definition 4.5 characterizes these situations—later, we obtain the minimum locus covers for all trees and grid graphs. It is worth noting that the minimum locus cover need not be unique for a fixed graph.

**Definition 4.6** (Identifying locus cover). Given a set $\Lambda \subseteq \mathcal{Y}$, if $\Lambda$ is a locus cover where $\forall y \in \mathcal{Y}$, $\exists \mathbf{w} \in \Delta^{|\Lambda|-1}$ such that $m_{\Lambda}(\mathbf{w}) = \{y\}$, then $\Lambda$ is an identifying locus cover.

The Fréchet mean need not be unique—as an $\arg \min$, it returns a set of minimizers. In certain metric spaces, the minimum locus cover can yield large sets of minimizers—this is undesirable, as it makes predicting a single class challenging. Definition 4.6 appeals to the idea of finding some set of classes for which the Fréchet mean *always* returns a unique minimizer—this is desirable in practice, and in some cases, moreso than Definition 4.5.

**Definition 4.7** (Pairwise decomposable). Given $\Lambda \subseteq \mathcal{Y}$, $\Pi(\Lambda)$ is called pairwise decomposable when it holds that $\Pi(\Lambda) = \cup_{\lambda_1, \lambda_2 \in \Lambda} \Pi(\{\lambda_1, \lambda_2\})$.

In many cases, the locus can be written in a more convenient form—the union of the locus of pairs of nodes. We refer to this definition as pairwise decomposability. Later, we shall see that pairwise decomposability is useful in computing the locus in polynomial time.

**Trees** Many label spaces are endowed with a tree metric in practice: hierarchical classification, in which the label space includes both classes and superclasses, partial labeling problems in which internal nodes can represent the prediction of a set of classes, and the approximation of complex or intractable metrics using a minimum spanning tree. We show that for our purposes, trees have certain desirable properties that make them easy to use with LOKI—namely that we can easily identify a locus cover that satisfies both Definition 4.5 and Definition 4.6. Conveniently, we also show that any locus in any tree satisfies Definition 4.7.

We first note that the leaves of any tree yield the minimum locus cover. This is a convenient property— any label from any label space endowed with a tree metric can be predicted using LOKI using only classifiers trained using labels corresponding to the leaves of the metric space. This can be especially useful if the tree has long branches and few leaves. Additionally, for tree metric spaces, the minimum locus cover (Definition 4.5) is also an identifying locus cover (Definition 4.6). This follows from the construction of the weights in the proof of Theorem A.4 (shown in the Appendix) and the property that all paths in trees are unique. Finally, we note that any locus in any tree is pairwise decomposable—the proof of this is given in the Appendix (Lemma A.5). We will see later that this property yields an efficient algorithm for computing the locus.

**Phylogenetic Trees** Image classification datasets often have a hierarchical tree structure, where only the leaves are actual classes, and internal nodes are designated as superclasses—examples include the ImageNet [6] and CIFAR-100 datasets [20]. Tree graphs in which only the leaf nodes are labeled are referred to as phylogenetic trees [3]. Often, these graphs are weighted, but unless otherwise mentioned, we assume that the graph is unweighted.

For any arbitrary tree $T = (\mathcal{V}, \mathcal{E})$, the set of labels induced by phylogenetic tree graph is $\mathcal{Y} =$ Leaves($T$). We provide a heuristic algorithm for obtaining locus covers for arbitrary phylogenetic trees in Algorithm B.1 (see Appendix). We prioritize adding endpoints of long paths to $\Lambda$, and continue adding nodes in this way until $\Pi(\Lambda) = \mathcal{Y}$. Similarly to tree metric spaces, any phylogenetic tree metric space is pairwise decomposable. We prove the correctness of Algorithm B.1 and pairwise

decomposability of phylogenetic trees in the Appendix (Theorem A.6 and Lemma A.7). Later, we give algorithms for computing the set of nodes in an arbitrary locus in arbitrary graphs—if the locus is pairwise decomposable, the algorithm for doing so is efficient, and if not, it has time complexity exponential in $K$. Due to the pairwise decomposability of phylogenetic trees, this polynomial-time algorithm to compute $\Pi(\Lambda)$ applies.

**Grid Graphs**  Classes often have a spatial relationship. For example, classification on maps or the discretization of a manifold both have spatial relationships—grid graphs are well suited to these types of spatial relationships. We obtain minimum locus covers for grid graphs satisfying Definition 4.5, but we find that these are not generally identifying locus covers. On the other hand, we give an example of a simple identifying locus cover satisfying Definition 4.6. Again, we find that grid graphs are in general pairwise decomposable and hence follow Definition 4.7.

We find that the pair of vertices on furthest opposite corners yields the minimum locus cover. While the set of vertices given by Theorem A.8 (found in the Appendix) satisfies Definition 4.5, this set does not in general satisfy Definition 4.6. This is because the path between any two vertices is not unique, so each minimum path of the same length between the pair of vertices can have an equivalent minimizer. On contrast, the following example set of vertices satisfies Definition 4.6 but it clearly does not satisfy Definition 4.5. *Example*: Given a grid graph, the set of all corners is an identifying locus cover. On the other hand, the vertices given by Theorem A.8 can be useful for other purposes. Lemma A.9 (provided in the Appendix) shows that subspaces of grid graphs can be formed by the loci of pairs of vertices in $\Lambda$. This in turn helps to show that loci in grid graphs are pairwise decomposable in general (see Lemma A.10 in the Appendix).

**The Complete Graph**  The standard classification setting does not use relational information between classes. As before, we model this setting using the complete graph, and we show the expected result that in the absence of useful relational information, LOKI cannot help, and the problem once again becomes standard multiclass classification among observed classes. To do so, we show that there is no nontrivial locus cover for the complete graph (Theorem A.11 in the Appendix).

### 4.3   Label Subspaces in Practice

While it is desirable for the set of observed classes to form a minimum or identifying locus cover, it is often not possible to choose the initial set of observed classes a priori—these are often random. In this section, we describe the more realistic cases in which a random set of classes are observed and an active learning-based strategy to choose the next observed class. The aim of our active learning approach is, instead of randomly selecting the next observed class, to actively select the next class so as to maximize the total size of the locus—i.e., the number of possible classes that can be output using LOKI. Before maximizing the locus via active learning, we must first address a much more basic question: can we even efficiently compute the locus?

**Computing the Locus**  We provide algorithms for obtaining the set of all classes in the locus, given a set of classes $\Lambda$. We show that when the locus is pairwise decomposable (Definition 4.7), we can compute the locus efficiently using a polynomial time algorithm. When the locus is not pairwise decomposable, we provide a general algorithm that has time complexity exponential in $|\Lambda|$—we are not aware of a more efficient algorithm. We note that any locus for every type of graph that we consider in Section 4.2 is pairwise decomposable, so our polynomial time algorithm applies. Algorithms B.2 and B.3 along with their time complexity analyses can be found in the Appendix.

**Large Locus via Active Next-Class Selection**  We now turn to actively selecting the next class to observe in order to maximize the size of the locus. For this analysis, we focus on the active learning setting when the class structure is a tree graph, as tree graphs are generic enough to apply to a wide variety of cases—including approximating other graphs using the minimum spanning tree. Assume the initial set of $K$ observed classes are sampled at random from some distribution. We would like to actively select the $K + 1$st class such that $|\Pi(\Lambda)|$ with $\Lambda = \{\lambda\}_{i=1}^{K+1}$ is as large as possible.

**Theorem 4.8.** *Let $T = (\mathcal{Y}, \mathcal{E})$ be a tree graph and let $\Lambda \subseteq \mathcal{Y}$ with $K = |\Lambda|$. Let $T'$ be the subgraph of the locus $\Pi(\Lambda)$. The vertex $v \in \mathcal{Y} \setminus \Lambda$ that maximizes $|\Pi(\Lambda \cup \{v\})|$ is the solution to the following optimization problem:* $\arg\max_{y \in \mathcal{Y} \setminus \Pi(\Lambda)} d(y, b)$ *s.t.* $b \in \partial_{in} T'$ *and* $\Gamma(y, b) \setminus \{b\} \subseteq \mathcal{Y} \setminus \Pi(\Lambda)$. *where $\partial_{in} T'$ is the inner boundary of $T'$ (all vertices in $T'$ that share an edge with vertices not in $T'$).*

Table 1: **CIFAR-100.** Improving CLIP predictions using LOKI. Results are reported as $\mathbb{E}[d^2(y, \hat{y})]$ in the respective metric space. CLIP-like zero-shot models can be improved using LOKI even without access to an external metric, and internal class embedding distances are used. When an external metric is available, LOKI outperforms CLIP using the default CIFAR-100 hierarchy and WordNet.

| Model | Metric Space | $\arg\max$ | LOKI | Relative Improvement |
|---|---|---|---|---|
| CLIP-RN50 [32] | Internal | 0.2922 | **0.2613** | **10.57%** |
| CLIP-ViT-L-14 [32] | Internal | 0.1588 | **0.1562** | **1.63%** |
| ALIGN [16] | Internal | 0.1475 | **0.1430** | **3.02%** |
| CLIP-RN50 [32] | $K_{100}$ | 0.5941* | 0.5941* | 0.0%* |
| CLIP-RN50 [32] | Default tree | 7.3528 | **7.1888** | **2.23%** |
| CLIP-RN50 [32] | WordNet tree | 24.3017 | **19.5549** | **19.53%** |

\* methods equivalent under the complete graph as LOKI reduces to $\arg\max$ prediction.

This procedure can be computed in polynomial time—solving the optimization problem in Theorem 4.8 simply requires searching over pairs of vertices. Hence we have provided an efficient active learning-based strategy to maximize the size of the locus for trees.

## 5 Experimental Results

In this section, we provide experimental results to validate the following claims:

1. LOKI improves performance of zero-shot foundation models even with no external metric.
2. LOKI adapts to label spaces with a large number of unobserved classes.
3. The active approach given in Theorem 4.8 yields a larger locus than the passive baseline.
4. The same active approach yields better performance on ImageNet.
5. With LOKI, calibration can improve *accuracy*, even with no external metric.

### 5.1 LOKI Improves Zero-Shot Models

We evaluate the capability of LOKI to improve upon zero-shot models where all classes are observed.

**Setup** Our experiment compares the zero-shot prediction performance of CLIP [32] on CIFAR-100 [20] to CLIP logits used with LOKI. First, we consider the setting in which no external metric relating the labels is available, and instead derive internal metric information from Euclidean distances between text embeddings from the models using their respective text encoders. Second, we consider three external metric spaces for use with LOKI: the complete graph $K_{100}$ and two phylogenetic trees: the default CIFAR-100 superclasses [20] and WordNet [2].

**Results** The results of this experiment are given in Table 1. When no external metric information is available, LOKI still outperforms CLIP-like models that use the standard prediction rule—in other words, LOKI seems to unconditionally improve CLIP. As expected, under the complete graph, our method becomes equivalent to the standard prediction mechanism used by CLIP. On the other hand, LOKI outperforms CLIP when using the default CIFAR-100 tree hierarchy and even more so when using the WordNet geometry, with a **19.53**% relative improvement in mean squared distance over the CLIP baseline. We postulate that the strong performance using WordNet is due to the richer geometric structure compared to that of the default CIFAR-100 hierarchy.

### 5.2 LOKI on Partially-Observed Label Spaces

To validate our approach on partially observed label spaces, we evaluate the performance of adapting a logistic classifier trained on SimCLR embeddings of ImageNet [6], 5-NN models trained on a 9,419 class subset of the PubMed dataset,[2] and the 325,056-class LSHTC dataset [30].

---

[2] https://www.kaggle.com/datasets/owaiskhan9654/pubmed-multilabel-text-classification

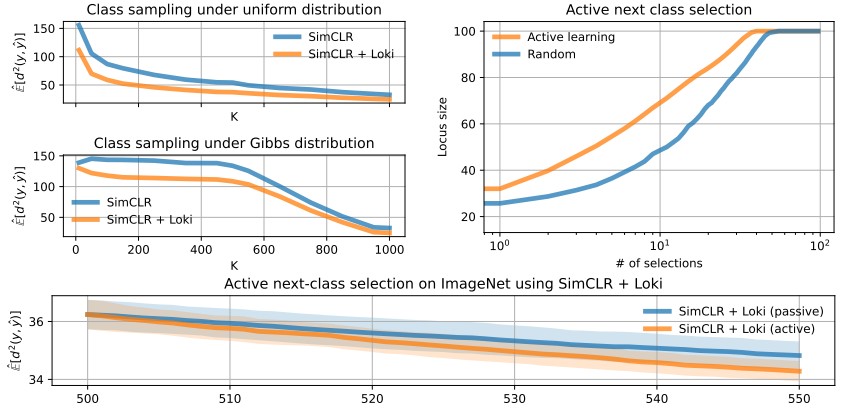

Figure 2: (Top left) **ImageNet.** Mean squared distances under uniform class sampling and under the Gibbs distribution—LOKI improves upon the baseline SimCLR one-vs-rest classifier. (Top right) **Synthetic.** Active class selection consistently leads to larger loci compared to uniform sampling. (Bottom) **Active selection on ImageNet.** Active class selection improves performance on ImageNet.

**Setup** For ImageNet, we use the WordNet phylogenetic tree as the metric space [2]. In this setting, we sample random subsets of size $K$ of the 1000 ImageNet classes and compare a baseline one-vs-rest classifier to the same classifier but using LOKI to adapt predictions to classes beyond the original $K$. We conduct two sets of experiments. In the first, we sample $K$ classes uniformly, while in the second, we adopt a more realistic sampler—we sample from a Gibbs distribution: $P(\lambda|\theta) = \frac{1}{Z}\exp(-\theta d(\lambda, \lambda^c))$, where $\lambda^c = m_{\mathcal{Y}}(\mathbf{1}_N)$ is the centroid of the metric space, $\theta$ is the concentration parameter around the centroid, and $Z$ is the normalizer. While the Gibbs distribution sampler is more realistic, it is also the more challenging setting—classes which have low probability according to this distribution are less likely to appear in the locus. For PubMed, we derive our metric from Euclidean distances between SimCSE class embeddings [11]. Finally for LSHTC, we summarize the default graph by randomly selecting nodes and merging them with their neighbors until we obtain a graph with 10,000 supernodes representing sets of classes.

**Results** Figure 2 shows the mean squared distances compared to the baseline one-vs-rest classifiers, across various settings of $K$. We find that LOKI always significantly outperforms the baseline, even in the more challenging setting of sampling according to a Gibbs distribution. Tables 2 and 3 show our improvements when using LOKI over the baseline 5-NN models. While LOKI consistently yields an improvement on PubMed and LSHTC, the improvement is more dramatic on LSHTC.

Table 2: **PubMed.** LOKI improves baseline for all settings of $K$. The metric space is Euclidean distances applied to SimCSE embeddings.

| $K$ | 5-NN | 5-NN + LOKI |
|---|---|---|
| 100 | 1.68666 | **1.42591** |
| 250 | 1.52374 | **1.47801** |
| 500 | 1.64074 | **1.45921** |

Table 3: **LSHTC.** LOKI improves baseline for all settings of $K$. We summarize the metric space graph by generating 10,000 supernodes.

| $K$ | 5-NN | 5-NN + LOKI |
|---|---|---|
| 50 | 1.2519 | **0.2896** |
| 100 | 1.4476 | **0.3467** |
| 150 | 1.7225 | **0.3969** |
| 200 | 1.7092 | **0.3171** |
| 250 | 1.6465 | **0.3995** |
| 300 | 1.6465 | **0.4004** |

## 5.3 Large Loci via Active Learning

We evaluate our active next class selection approach on increasing the locus size. We expect that compared to passive (random) selection, our active approach will lead to larger loci, and in practice, a larger set of possible classes that can be predicted using LOKI while observing fewer classes during training.

**Setup** We compare with a baseline that randomly selects the next class. We first generate a synthetic random tree with size $N$ and fix an initial $K$. In active selection, we use the approach described in Theorem 4.8 to select the next node. As a passive baseline, we randomly sample (without replacement) the remaining nodes that are not in $\Lambda$.

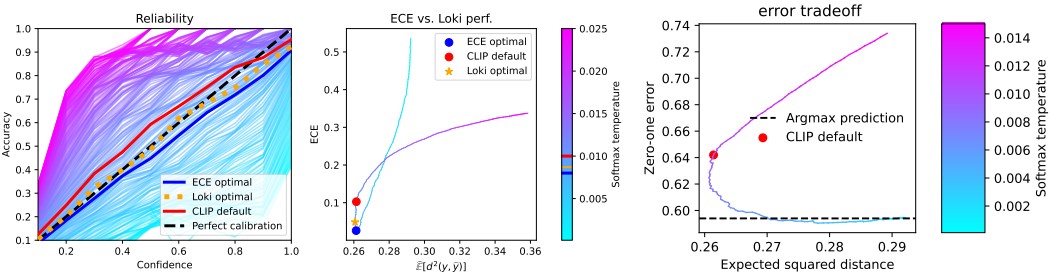

Figure 3: **CLIP on CIFAR-100 with no external metric.** (Left) Reliability diagrams across a range of Softmax temperatures, highlighting the CLIP default temperature, the optimal temperature for LOKI, and the minimizer of the Expected Calibration Error (ECE). All three are well-calibrated. (Center) Tradeoff between optimizing for ECE and the expected squared distance. As with the reliability diagrams, the CLIP default temperature, the LOKI-optimal temperature, and the ECE-optimal temperature are similar. (Right) Tradeoff between optimizing for zero-one error and the expected squared distance. Temperature can be tuned to improve *accuracy* when using LOKI.

**Results** The results are shown in Figure 2. We set $N = 100$ and the initial $K = 3$, then we average the result over 10 independent trials. The active approach consistently outperforms the random baseline as it attains a larger locus with fewer nodes selected.

## 5.4 Improving Performance via Active Learning

Next, we evaluate the our active next class selection approach on improving error on ImageNet. While we found that the the active approach indeed leads to an increased locus size compared to the passive baseline, we expect that this increased locus size will lead to improved performance.

**Setup** We randomly sample 500 ImageNet classes, and using the WordNet metric space, we use our active approach to iteratively sample 50 more classes. We compare this to a passive baseline in which the 50 classes are sampled randomly. We repeat this experiment over 10 independent trials.

**Results** Figure 2 shows that our active approach yields improved error over the passive baseline. The gap between the active approach and the passive baseline widens with more selection rounds.

## 5.5 Improving Accuracy via Calibration

Finally, we evaluate the effect of calibrating the Softmax outputs on the performance of LOKI.

**Setup** We calibrate via Softmax temperature scaling [13] using CLIP on CIFAR-100. We do not use an external metric space, and instead use Euclidean distance applied to the CLIP text encoder.

**Results** The reliability diagram in Figure 3 shows that the optimal Softmax temperature for LOKI is both close to the default temperature used by CLIP and to the optimally-calibrated temperature. In Figure 3 (right), we find that appropriate tuning of the temperature parameter *can lead to improved accuracy with CLIP*, even when no external metric space is available.

# 6 Conclusion

In this work, we proposed LOKI—a simple adaptor for pretrained models to enable the prediction of additional classes that are unobserved during training by using metric space information. We comprehensively answered the space of new questions that arise under LOKI in terms of learning, optimal metric space settings, and a practical active selection strategy. Experimentally, we showed that LOKI can be used to improve CLIP even without external metric space information, can be used to predict a large number of unobserved classes, and a validation of our active selection strategy.

## Acknowledgements

We are grateful for the support of the NSF under CCF2106707 (Program Synthesis for Weak Supervision) and the Wisconsin Alumni Research Foundation (WARF).

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

# A    Deferred Proofs

## A.1    Proof of Theorem 4.1 (LOKI sample complexity)

In this section, we provide a formal statement and proof of our sample complexity result for LOKI, including additional required definitions and assumptions.

First, we define the required tools to prove the sample complexity bound for LOKI. For the purposes of this proof, we define the Fréchet variance as the function over which the Fréchet mean returns the minimizer.

**Definition A.1** (Fréchet variance). The Fréchet variance is defined as

$$\Psi_{\mathbf{w}}(V) := \sum_{i \in [K]} \mathbf{w}_i d^2(V, V_i).$$

Additionally, we will require a technical assumption related to the sensitivity of the Fréchet variance to different node choices.

**Assumption A.2** (Fréchet variance is $\frac{1}{\alpha}$-bi-Lipschitz). For a metric space defined by a graph $G = (\mathcal{V}, \mathcal{E})$, the Fréchet variance is $K$-bi-Lipschitz if there exists a $K \geq 1$ such that

$$\frac{1}{K}d^2(V, \tilde{V}) \leq |\Psi_{\mathbf{w}}(V) - \Psi_{\mathbf{w}}(\tilde{V})| \leq K d^2(V, \tilde{V})$$

for all $V, \tilde{V} \in \mathcal{V}$, and a fixed $\mathbf{w} \in \Delta^{K-1}$. For our purposes, such a $K$ always exists: consider setting $K = \text{diam}(G)^2 \max_{V_1, V_2 \in \mathcal{V}} |\Psi_{\mathbf{w}}(V_1) - \Psi_{\mathbf{w}}(V_2)|$. However, this is a very conservative bound that holds for all graphs that we consider. Instead, we assume access to the largest $\alpha = \frac{1}{K} \leq 1$ such that

$$\alpha d^2(V, \tilde{V}) \leq |\Psi_{\mathbf{w}}(V) - \Psi_{\mathbf{w}}(\tilde{V})|,$$

which may be problem dependent.

**Theorem A.3** (LOKI sample complexity). *Let $\mathcal{Y}$ be a set of points on the $d$ dimensional 2-norm ball of radius $R$, and let $\Lambda \subseteq \mathcal{Y}$ be the set of $K$ observed classes. Assume that $\Lambda$ forms a $2R/(\sqrt{d}K - 1)$-net under the Euclidean distance. Assume that training examples are generated by drawing $n$ samples from the following process: draw $x \sim \mathcal{N}(y, I)$ where $y \sim \text{Unif}(\Lambda)$, and at test time, draw $x \sim \mathcal{N}(y, I)$ where $y \sim \text{Unif}(\mathcal{Y})$. Assume that we estimate a Gaussian mixture model with $K$ components (each having identity covariance) on the training set and obtain probability estimates $\hat{\mathbb{P}}(y_i|x)$ for $i \in [K]$ for a sample $(x, y_*)$ from the test distribution. Then with high probability, under the following model,*

$$\hat{y}_* \in m_\Lambda([\hat{\mathbb{P}}(y_i|x)]_{i \in [K]}) = \arg\min_{y \in \mathcal{Y}} \sum_{i \in [K]} \hat{\mathbb{P}}(y_i|x) d^2(y, y_i)$$

*the sample complexity of estimating target $y_*$ from the test distribution $\mathcal{D}_{test}$ with prediction $\hat{y}_*$ is:*

$$\mathbb{E}_{(x, y_*) \sim \mathcal{D}_{test}}[d^2(y_*, \hat{y}_*)] \leq O\left(\frac{d}{\alpha}\sqrt{\frac{\log K/\delta}{n}}\left(\frac{1}{\left(R^{1-\frac{2}{d}} - \frac{\log R}{R}\right)} + \sqrt{d}\right)\right)$$

*where $d$ is the dimensionality of the input and $\alpha$ is our parameter under Assumption A.2.*

*Proof.* We begin by detailing the data-generating process.

**At training time,** our underlying data-generating process, $\mathcal{D}_{\text{train}}$, is as follows:

- We begin with a $\frac{2R}{\sqrt{d}K-1}$-net, $\mathcal{Y}$, on the $d$ dimensional 2-norm ball of radius $R$ under the Euclidean distance, and let $\Lambda \subseteq \mathcal{Y}$

- Draw $y \sim \text{Unif}(\mathcal{Y})$.

- Discard draws if $y \notin \Lambda$.

- Draw $x \sim \mathcal{N}(y, I)$.

**At test time,** we do not discard draws and allow for classes not in $\Lambda$ and use the following data generating process, $\mathcal{D}_{\text{test}}$:

- We begin with a $\frac{2R}{\sqrt{d}K-1}$-net, $\mathcal{Y}$, on the $d$ dimensional 2-norm ball of radius $R$ under the Euclidean distance, and let $\Lambda \subseteq \mathcal{Y}$

- Draw $y \sim \text{Unif}(\mathcal{Y})$.

- Draw $x \sim \mathcal{N}(y, I)$.

Given a labeled training dataset $D = \{(x_i, y_i)\}_{i=1}^n$ containing $n$ points drawn from $\mathcal{D}_{\text{train}}$ with $|\Lambda| = K$ distinct classes, we would like to fit a $K$-component Gaussian mixture model with identity covariance. We first perform mean estimation of each of the classes separately using the median-of-means estimator [15, 27, 24]. Using the estimator yields the following parameter estimation bound [27, 4]:

$$||y_i - \hat{y_i}||_2 \leq O\left(\sqrt{\frac{d \log K/\delta}{n}}\right)$$

with probability $1 - \delta$.

Next, we consider the relationships between four quantities: $\Psi_\mathbb{P}(y_*)$, $\Psi_\mathbb{P}(\hat{y}_*)$, $\Psi_{\widehat{\mathbb{P}}}(y_*)$, $\Psi_{\widehat{\mathbb{P}}}(\hat{y}_*)$, where $\mathbb{P}$ is the vector of probabilities from the true Gaussian mixture, $\widehat{\mathbb{P}}$ is the vector of probabilities from the estimated model, $y_* \in m_\Lambda(\mathbb{P})$ is the target class, and $\hat{y}_* \in m_\Lambda(\widehat{\mathbb{P}})$ is the predicted class. While it is problem-dependent as to whether $\Psi_\mathbb{P}(y_*) \leq \Psi_{\widehat{\mathbb{P}}}(\hat{y}_*)$ or $\Psi_\mathbb{P}(y_*) > \Psi_{\widehat{\mathbb{P}}}(\hat{y}_*)$, a similar argument holds for both cases. So without loss of generality, we assume that $\Psi_\mathbb{P}(y_*) \leq \Psi_{\widehat{\mathbb{P}}}(\hat{y}_*)$. Then by the definition of the Fréchet mean, the following inequalities hold:

$$\Psi_\mathbb{P}(y_*) \leq \Psi_{\widehat{\mathbb{P}}}(\hat{y}_*) \leq \Psi_{\widehat{\mathbb{P}}}(y_*),$$

and consequently,

$$\Psi_{\widehat{\mathbb{P}}}(\hat{y}_*) - \Psi_\mathbb{P}(y_*) \leq \Psi_{\widehat{\mathbb{P}}}(y_*) - \Psi_\mathbb{P}(y_*). \tag{2}$$

We proceed by obtaining upper and lower bounds of the existing bounds in Equation 2. First, we will obtain an upper bound on $\Psi_{\widehat{\mathbb{P}}}(y_*) - \Psi_\mathbb{P}(y_*)$.

$$
\begin{aligned}
|\Psi_{\widehat{\mathbb{P}}}(y_*) - \Psi_\mathbb{P}(y_*)| &= \left| \sum_{i \in [K]} (\widehat{\mathbb{P}} - \mathbb{P}) d^2(y_*, V_i) \right| \\
&= \left| \sum_{i \in [K]} \left( \widehat{\mathbb{P}}(y_i|x) - \mathbb{P}(y_i|x) \right) ||y_* - y_i||_2^2 \right| \\
&= \left| \sum_{i \in [K]} \left( \frac{\widehat{\mathbb{P}}(x|y_i)\mathbb{P}(y_i)}{\sum_{j \in [K]} \widehat{\mathbb{P}}(x|y_j)\mathbb{P}(y_j)} - \frac{\mathbb{P}(x|y_i)\mathbb{P}(y_i)}{\sum_{j \in [K]} \mathbb{P}(x|y_j)\mathbb{P}(y_j)} \right) ||y_* - y_i||_2^2 \right|
\end{aligned}
\tag{3}
$$

$$= \left| \sum_{i \in [K]} \left( \frac{\exp\{-\frac{1}{2}||x - \hat{y}_i||_2^2\} \frac{1}{K}}{\sum_{j \in [K]} \exp\{-\frac{1}{2}||x - \hat{y}_j||_2^2\} \frac{1}{K}} \right. \right.$$

$$\left. \left. - \frac{\exp\{-\frac{1}{2}||x - y_i||_2^2\} \frac{1}{K}}{\sum_{j \in [K]} \exp\{-\frac{1}{2}||x - y_j||_2^2\} \frac{1}{K}} \right) ||y_* - y_i||_2^2 \right|.$$

$$= \left| \sum_{i \in [K]} \left( \frac{\exp\{-\frac{1}{2}||x - \hat{y}_i||_2^2\}}{\sum_{j \in [K]} \exp\{-\frac{1}{2}||x - \hat{y}_j||_2^2\}} \right. \right.$$

$$\left. \left. - \frac{\exp\{-\frac{1}{2}||x - y_i||_2^2\}}{\sum_{j \in [K]} \exp\{-\frac{1}{2}||x - y_j||_2^2\}} \right) ||y_* - y_i||_2^2 \right|. \tag{4}$$

For notational convenience, we define the following:

$a_i := \exp\{-\frac{1}{2}||x - y_i||_2^2\}$,
$\hat{a}_i := \exp\{-\frac{1}{2}||x - \hat{y}_i||_2^2\}$,
$b := \sum_{j \in [K]} \exp\{-\frac{1}{2}||x - y_j||_2^2\}$,
$\hat{b} := \sum_{j \in [K]} \exp\{-\frac{1}{2}||x - \hat{y}_j||_2^2\}$, and
$c_i := ||\hat{y}_* - y_i||_2^2$.

Then (4) becomes:

$$\left| \sum_{i \in [K]} \left( \frac{\hat{a}_i}{\hat{b}} - \frac{a_i}{b} \right) c_i \right| = \left| \sum_{i \in [K]} \left( \frac{\hat{a}_i}{\hat{b}} - \frac{\hat{a}_i}{b} + \frac{\hat{a}_i}{b} - \frac{a_i}{b} \right) c_i \right|$$

$$= \left| \sum_{i \in [K]} \left( \frac{\hat{a}_i - a_i}{b} + \hat{a}_i \left( \frac{1}{\hat{b}} - \frac{1}{b} \right) \right) c_i \right|$$

$$\leq \left| \sum_{i \in [K]} \left( \frac{\hat{a}_i - a_i}{b} \right) c_i \right| + \left| \sum_{i \in [K]} \left( \hat{a}_i \left( \frac{1}{\hat{b}} - \frac{1}{b} \right) \right) c_i \right|$$

$$= \left| \sum_{i \in [K]} \frac{a_i}{b} \left( \frac{\hat{a}_i}{a_i} - 1 \right) c_i \right| + \left| \frac{b - \hat{b}}{b} \sum_{i \in [K]} \frac{\hat{a}_i}{\hat{b}} c_i \right|$$

$$= \left| \sum_{i \in [K]} \frac{a_i}{b} \left( \frac{\hat{a}_i}{a_i} - 1 \right) c_i \right| + \left| \left( \sum_{i \in [K]} \frac{a_i}{b} \left( \frac{\hat{a}_i}{a_i} - 1 \right) \right) \left( \sum_{i \in [K]} \frac{\hat{a}_i}{\hat{b}} c_i \right) \right|. \tag{5}$$

Now we define the following: $L := ||x - y_z||_2^2$ with $z \in \arg\min_j ||x - y_j||_2^2$ is the smallest distance to a class mean, and $L + E_i := ||x - y_i||_2^2$ with $E_i > 0$. Similarly, define $\hat{L} := ||x - \hat{y}_z||_2^2$ with $z \in \arg\min_j ||x - \hat{y}_j||_2^2$ is the smallest distance to an estimated class mean, and $\hat{L} + \hat{E}_i := ||x - \hat{y}_i||_2^2$ with $\hat{E}_i > 0$. Finally, we define $T := \sqrt{\hat{L}} + \sqrt{\hat{L} + \hat{E}_i} + \sqrt{L} + \sqrt{L + E_i}$.

Then we can bound the parts separately:

For $i \neq z$, we have

$$\frac{a_i}{b} = \left( \frac{\exp\{-\frac{1}{2}||x - y_i||_2^2\}}{\sum_{j \in [K]} \exp\{-\frac{1}{2}||x - y_j||_2^2\}} \right)$$

$$\leq \left( \frac{\exp\{-\frac{1}{2}(L + E_i)\}}{\exp\{-\frac{1}{2}L\} + \exp\{-\frac{1}{2}(L + E_i)\}} \right)$$

$$= \frac{1}{\exp\{\frac{1}{2}E_i\}} \tag{6}$$

and in the case of $i = z$, we bound $\frac{a_z}{b} \leq 1$. So overall, we have $\frac{a_i}{b} \leq \frac{1}{\exp\{\mathbf{1}_{i \neq z} \frac{1}{2}E_i\}}$ for all $i \in [K]$.

$$c_i = ||y_* - y_i||_2^2$$
$$\leq 2||x - y_i||_2^2 + 2||x - y_*||_2^2$$
$$= 2(L + E_i + ||x - y_*||_2^2). \tag{7}$$

$$\frac{\hat{a}_i}{a_i} - 1 = \exp\left\{\frac{1}{2}||x - y_i||_2^2 - \frac{1}{2}||x - \hat{y}_i||_2^2\right\} - 1$$
$$\approx 1 + \frac{1}{2}||x - y_i||_2^2 - \frac{1}{2}||x - \hat{y}_i||_2^2 - 1$$
$$= \frac{1}{2}\langle \hat{y}_i - y_i, 2x - y_i - \hat{y}_i \rangle$$
$$\tag{8}$$

$$\leq \frac{1}{2}||\hat{y}_i - y_i|| \cdot ||2x - y_i - \hat{y}_i||$$
$$\leq ||\hat{y}_i - y_i|| \left(||x - y_i|| + ||x - \hat{y}_i||\right)$$
$$= ||\hat{y}_i - y_i|| \left(||x - y_i|| + ||x - y_i + y_i - \hat{y}_i||\right)$$
$$\leq ||\hat{y}_i - y_i|| \left(2||x - y_i|| + ||y_i - \hat{y}_i||\right)$$
$$= ||\hat{y}_i - y_i|| \left(2\sqrt{L + E_i} + ||y_i - \hat{y}_i||\right). \tag{9}$$

Next, we must control $E_i - \hat{E}_i$ in order to obtain the bound for $\frac{\hat{a}_i}{\hat{b}}$.

$$E_i - \hat{E}_i = (||x - y_i||^2 - L) - (||x - \hat{y}_i||^2 - \hat{L})$$
$$= ||x - y_i||^2 - ||x - \hat{y}_i||^2 + ||x - \hat{y}_z||^2 - ||x - y_z||^2$$
$$= \langle \hat{y}_i - y_i, 2x - y_i - \hat{y}_i \rangle + \langle y_z - \hat{y}_z, 2x - \hat{y}_z - y_z \rangle$$
$$\leq ||\hat{y}_i - y_i|| \cdot ||2x - y_i - \hat{y}_i|| + ||y_z - \hat{y}_z|| \cdot ||2x - \hat{y}_z - y_z||$$
$$\leq ||y_i - \hat{y}_i|| \left(||x - \hat{y}_i|| + ||x - y_i||\right) + ||\hat{y}_z - y_z|| \left(||x - y_z|| + ||x - \hat{y}_z||\right)$$
$$\leq c\sqrt{\frac{d\log K/\delta}{n}} \left(||x - \hat{y}_i|| + ||x - y_i|| + ||x - y_z|| + ||x - \hat{y}_z||\right)$$
$$= c\sqrt{\frac{d\log K/\delta}{n}} \left(\sqrt{\hat{L}} + \sqrt{\hat{L} + \hat{E}_i} + \sqrt{L} + \sqrt{L + E_i}\right)$$
$$\hat{E}_i \geq \max\left\{0, E_i - cT\sqrt{\frac{d\log K/\delta}{n}}\right\}.$$

Using (6), we obtain

$$\frac{\hat{a}_i}{\hat{b}} \leq \frac{1}{\exp\{\mathbf{1}_{i\neq z}\frac{1}{2}\hat{E}_i\}}$$
$$\leq \frac{1}{\exp\left\{\mathbf{1}_{i\neq z}\max\left\{0, \frac{1}{2}E_i - \frac{cT}{2}\sqrt{\frac{d\log K/\delta}{n}}\right\}\right\}} \tag{10}$$

Plugging (6), (7), (9), and (10) into (5), we obtain

$$\left|\sum_{i\in[K]} \frac{a_i}{b}\left(\frac{\hat{a}_i}{a_i} - 1\right)c_i\right| + \left|\left(\sum_{i\in[K]} \frac{a_i}{b}\left(\frac{\hat{a}_i}{a_i} - 1\right)\right)\left(\sum_{i\in[K]} \frac{\hat{a}_i}{\hat{b}}c_i\right)\right|$$

$$\leq \sum_{i\in[K]} \frac{||\hat{y}_i - y_i||\left(2\sqrt{L+E_i} + ||y_i - \hat{y}_i||\right)}{\exp\{\mathbf{1}_{i\neq z}\frac{1}{2}E_i\}} 2(L + E_i + ||x - y_*||_2^2)$$

$$+ \left( \sum_{i\in[K]} \frac{||\hat{y}_i - y_i||\left(2\sqrt{L+E_i} + ||y_i - \hat{y}_i||\right)}{\exp\{\mathbf{1}_{i\neq z}\frac{1}{2}E_i\}} \right)$$

$$\cdot \left( \sum_{i\in[K]} \frac{2(L + E_i + ||x - y_*||_2^2)}{\exp\left\{\mathbf{1}_{i\neq z}\max\left\{0, \frac{1}{2}E_i - \frac{cT}{2}\sqrt{\frac{d\log K/\delta}{n}}\right\}\right\}} \right)$$

$$\leq O\left( \sqrt{\frac{d\log K/\delta}{n}} \sum_{i\in[K]} \frac{L + E_i + ||x - y_*||_2^2}{\exp\left\{\mathbf{1}_{i\neq z}\frac{1}{2}E_i\right\}} \right)$$

Next, recalling the fact that our class means form an $\varepsilon$-net on the radius-$R$ ball, we use Lemma 5.2 from [40] to bound $L$ as $L \leq \frac{2R}{\sqrt{d}K-1}$.

$$\leq O\left( \sqrt{\frac{d\log K/\delta}{n}} \sum_{i\in[K]} \frac{\frac{R}{\sqrt{d}K} + E_i + ||x - y_*||_2^2}{\exp\left\{\mathbf{1}_{i\neq z}\frac{1}{2}E_i\right\}} \right). \tag{11}$$

Next, we will consider cases on $E_i$. We will consider the cases in which $E_i < \log R^2$ and $E_i \geq \log R^2$.

We will begin with the case in which $E_i < \log R^2$, then Equation 11 becomes:

$$\leq O\left( \sqrt{\frac{d\log K/\delta}{n}} \left( \sum_{i\in[K]} \frac{R}{\sqrt{d}K} + ||x - y_*||_2^2 \right) \right).$$

Next, the case in which $E_i \geq \log R^2$, then Equation 11 becomes:

$$\leq O\left( \sqrt{\frac{d\log K/\delta}{n}} \sum_{i\in[K]} \frac{\frac{R}{\sqrt{d}K} + 2\log R + ||x - y_*||_2^2}{R} \right)$$

$$\leq O\left( \sqrt{\frac{d\log K/\delta}{n}} \left( \sum_{i\in[K]} \frac{1}{\sqrt{d}K} + ||x - y_*||_2^2 \right) \right).$$

Setting $M \leq K$ to be the number of terms for which $E_i < \log R^2$ holds and by combining the two bounds, we obtain the following:

$$\leq O\left( \sqrt{\frac{d\log K/\delta}{n}} \left( \left( \frac{MR}{\sqrt{d}K} + \frac{K-M}{\sqrt{d}K} \right) + K||x - y_*||_2^2 \right) \right).$$

Ultimately, our bound is

$$|\Psi_{\widehat{\mathbb{P}}}(y_*) - \Psi_{\mathbb{P}}(y_*)| \leq O\left( \sqrt{\frac{d\log K/\delta}{n}} \left( \left( \frac{MR}{\sqrt{d}K} + \frac{K-M}{\sqrt{d}K} \right) + K||x - y_*||_2^2 \right) \right). \tag{12}$$

Next, we will obtain a lower bound on $\Psi_{\widehat{\mathbb{P}}}(\hat{y}_*) - \Psi_{\mathbb{P}}(y_*)$.

$$|\Psi_{\widehat{\mathbb{P}}}(\hat{y}_*) - \Psi_{\mathbb{P}}(y_*)| \geq |\Psi_{\mathbb{P}}(y_*) - \Psi_{\mathbb{P}}(\hat{y}_*)| - |\Psi_{\widehat{\mathbb{P}}}(\hat{y}_*) - \Psi_{\mathbb{P}}(\hat{y}_*)| \quad \text{triangle inequality}$$
$$\geq \alpha d^2(y_*, \hat{y}_*) - |\Psi_{\widehat{\mathbb{P}}}(y_*) - \Psi_{\mathbb{P}}(y_*)| \quad \text{Assumption A.2.}$$

Combining both of these bounds with Equation 2, we obtain

$$\alpha d^2(y_*,\hat{y}_*) - |\Psi_{\widehat{\mathbb{P}}}(y_*) - \Psi_{\mathbb{P}}(y_*)| \leq \Psi_{\widehat{\mathbb{P}}}(\hat{y}_*) - \Psi_{\mathbb{P}}(y_*) \leq \Psi_{\widehat{\mathbb{P}}}(y_*) - \Psi_{\mathbb{P}}(y_*)$$
$$\leq |\Psi_{\widehat{\mathbb{P}}}(y_*) - \Psi_{\mathbb{P}}(y_*)|$$

$$\Rightarrow \alpha d^2(y_*,\hat{y}_*) \leq 2|\Psi_{\widehat{\mathbb{P}}}(y_*) - \Psi_{\mathbb{P}}(y_*)|$$

$$\Rightarrow d^2(y_*,\hat{y}_*) \leq O\left(\frac{1}{\alpha}\sqrt{\frac{d\log K/\delta}{n}}\left(\frac{MR}{\sqrt{dK}} + \frac{K-M}{\sqrt{dK}} + K\|x - y_*\|_2^2\right)\right)$$

$$\Rightarrow \mathbb{E}_{(x,y_*)\sim\mathcal{D}_{\text{test}}}[d^2(y_*,\hat{y}_*)] \leq O\left(\frac{1}{\alpha}\sqrt{\frac{d\log K/\delta}{n}}\left(\frac{MR}{\sqrt{dK}} + \frac{K-M}{\sqrt{dK}} + dK\right)\right),$$

Next, we obtain a bound on $M$. Since $M \leq K$ is the number of terms for which $E_i < \log R^2$, we have that $M = VK$, where $V \in [0,1]$ is the ratio of the volumes of the ball for which $E_i < \log R^2$, and the ball containing the entire set of classes $\mathcal{Y}$. We have

$$E_i = \|x - y_i\|_2^2 - \|x - y_z\|_2^2 < \log R^2$$
$$\|x - y_i\|_2^2 < \log R^2 + L$$
$$\leq \log R^2 + \frac{2R}{\sqrt{dK} - 1} \leq R^2$$
$$\Rightarrow \sqrt{\log R^2 + \frac{2R}{\sqrt{dK} - 1}} \leq R,$$

which is an upper bound of the radius of the ball for which $E_i < \log R^2$ is true. Now we construct $V$:

$$V = \left(\frac{\sqrt{\log R^2 + \frac{2R}{\sqrt{dK}-1}}}{R}\right)^d.$$

Combining this with our error bound, we have

$$\mathbb{E}_{(x,y_*)\sim\mathcal{D}_{\text{test}}}[d^2(y_*,\hat{y}_*)] \leq$$

$$O\left(\frac{1}{\alpha}\sqrt{\frac{d\log K/\delta}{n}}\left(\frac{\left(\frac{\sqrt{\log R^2 + \frac{2R}{\sqrt{dK}-1}}}{R}\right)^d R}{\sqrt{d}} + \frac{1 - \left(\frac{\sqrt{\log R^2 + \frac{2R}{\sqrt{dK}-1}}}{R}\right)^d}{\sqrt{d}} + dK\right)\right).$$

However, we can choose $K$ to weaken our dependence on $R$.

$$V = \left(\frac{\sqrt{\log R^2 + \frac{2R}{\sqrt{dK}-1}}}{R}\right)^d \leq \frac{1}{R}$$

$$\Rightarrow K \geq \frac{2}{\sqrt{d}\left(R^{1-\frac{2}{d}} - \frac{\log R}{R}\right)} + 1.$$

Finally, we have

$$\mathbb{E}_{(x,y_*)\sim\mathcal{D}_{\text{test}}}[d^2(y_*,\hat{y}_*)] \leq O\left(\frac{1}{\alpha}\sqrt{\frac{d\log K/\delta}{n}}\left(\frac{1}{\sqrt{d}} + \frac{1 - \frac{1}{R}}{\sqrt{d}} + \frac{\sqrt{d}}{\left(R^{1-\frac{2}{d}} - \frac{\log R}{R}\right)} + d\right)\right)$$

$$\leq O\left(\frac{1}{\alpha}\sqrt{\frac{d\log K/\delta}{n}}\left(\frac{1}{\sqrt{d}} + \frac{\sqrt{d}}{\left(R^{1-\frac{2}{d}} - \frac{\log R}{R}\right)} + d\right)\right)$$

$$\leq O\left(\frac{d}{\alpha}\sqrt{\frac{\log K/\delta}{n}}\left(\frac{1}{\left(R^{1-\frac{2}{d}} - \frac{\log R}{R}\right)} + \sqrt{d}\right)\right)$$

which completes the proof. □

**Discussion of Theorem A.3.** The above bound contains several standard quantities, including the dimensionality of the inputs $d$, the radius $R$ of the ball from which labels are drawn, the number of classes and samples used to estimate the model $K$ and $n$ respectively, and $\delta$, which is used to control the probability that the bound holds. Naturally, the bound scales up with higher $d$ and $K$, and improves with an increased number of samples $n$. The dependence on $R$ is also related to its dependence on $d$ — so long as we have that $K \geq 2(\sqrt{d}(R^{1-\frac{2}{d}} - \log R/R))^{-1} + 1$ and for $d \geq 2$, $R$ increases, which improves the bound. The bound includes a metric space dependent quantity $\alpha$, which is related to the amount that the Fréchet variance can change subject to changes in its argument, i.e., which node is considered.

## A.2 Proof of Theorem A.4 (minimum locus cover for trees)

**Theorem A.4** (Minimum locus cover for trees)**.** *The minimum locus cover for a tree graph $T$ is $\Lambda = Leaves(T)$.*

*Proof.* We would like to show to show two things:

1. any node can be resolved by an appropriate construction of $\boldsymbol{w}$ using only the leaves and

2. removing any leaf makes that leaf node unreachable, no matter what the other nodes are in $\boldsymbol{y}$–i.e., we must use at least all of the leaf nodes.

If both of these conditions hold (i.e., that the leaves form a locus cover and that they are the smallest such set), then the the leaves of any tree form a minimum locus cover. To set up the problem, we begin with some undirected tree, $T$, whose nodes are our set of labels: $T = (\mathcal{Y}, E)$ where $\Lambda = \text{Leaves}(T) \subseteq \mathcal{Y}$ is the set of leaf nodes. Moreover, let $\boldsymbol{\Lambda}$ be a tuple of the leaf nodes. We will start by proving that $\Lambda$ is a locus cover. We will show this by cases on $v \in \mathcal{Y}$, the node that we would like to resolve:

1. If $v \in \Lambda$, i.e. $v$ is a leaf node, then setting $\boldsymbol{w}_i = \mathbf{1}\{\boldsymbol{\Lambda}_i = v\}$ yields

$$m_{\boldsymbol{\Lambda}}(\boldsymbol{w}) = \arg\min_{y \in \mathcal{Y}} \sum_{i=1}^{K} \mathbf{1}\{\boldsymbol{\Lambda}_i = v\}d^2(y, \boldsymbol{\Lambda}_i)$$

$$= \arg\min_{y \in \mathcal{Y}} d^2(y, v) = \{v\}.$$

2. If $v \notin \Lambda$, we have a bit more work to do. Since $v$ is an internal node, consider any pair of leaves, $l_1 \neq l_2$ such that $v$ is along the (unique) path between $l_1$ and $l_2$: $v \in \Gamma(l_1, l_2)$. Then

$$\text{set } \boldsymbol{w}_i = \begin{cases} \frac{d(v, l_2)}{d(l_1, l_2)} & \text{if } \boldsymbol{\Lambda_i} = l_1, \\ \frac{d(v, l_1)}{d(l_1, l_2)} & \text{if } \boldsymbol{\Lambda_i} = l_2, \\ 0 & \text{otherwise} \end{cases} \quad \text{yields}$$

$$m_{\boldsymbol{\Lambda}}(\boldsymbol{w}) = \arg\min_{y \in \mathcal{Y}} \frac{d(v, l_2)d^2(y, l_1) + d(v, l_1)d^2(y, l_2)}{d(l_1, l_2)}$$

$$= \arg\min_{y \in \mathcal{Y}} d(v, l_2)d^2(y, l_1) + d(v, l_1)d^2(y, l_2)$$

$$= \{v\}.$$

Thus $\Lambda$ is a locus cover. Next, we will show that it is the smallest such set. Let $l \in \Lambda$ be any leaf node, and define $\Lambda' = \Lambda \setminus \{l\}$. We must show that $\Lambda'$ is not a locus cover. Assume for contradiction that $\Lambda'$ is a locus cover. This means that given some tuple $\boldsymbol{y}'$ whose entries are the elements of $\Lambda'$, we have $\Pi(\Lambda') = \mathcal{Y}$. This implies that the path between any two nodes in $\Lambda$ is also contained in $\Pi(\boldsymbol{\Lambda}')$. Since the leaves form a locus cover and any $y \in \mathcal{Y}$ can be constructed by the appropriate choice of $\boldsymbol{w}$ along with a path between two leaf nodes, $l$ must either be one of the entries of $\boldsymbol{\Lambda}'$ (one of the endpoints of the path) or it must be an internal node. Both cases are contradictions–$\boldsymbol{\Lambda}'$ cannot include $l$ by assumption, and $l$ is assumed to be a leaf node. It follows that $\Lambda$ is a minimum locus cover. □

## A.3  Proof of Lemma A.5 (tree pairwise decomposability)

**Lemma A.5** (Tree pairwise decomposability)**.** *Let $T = (\mathcal{Y}, \mathcal{E})$ be a tree and $\Lambda \subseteq \mathcal{Y}$. Then $\Pi(\Lambda)$ is pairwise decomposable.*

*Proof.* Assume for contradiction that $\exists y^* \in \mathcal{Y}$ where $y^* \notin \Pi(\{\lambda_i, \lambda_j\}), \forall \lambda_i, \lambda_j \in \Lambda$, but $y^* \in \Pi(\Lambda)$. Then, $y^* \in m_\Lambda(\mathbf{w})$ for some $\mathbf{w}$. Note that $y^* \notin \Pi(\{\lambda_i, \lambda_j\})$ implies $y^* \notin \Gamma(\lambda_i, \lambda_j), \forall \lambda_i, \lambda_j \in \Lambda$, because $\Pi(\{\lambda_i, \lambda_j\}) = \Gamma(\lambda_i, \lambda_j)$ due to the uniqueness of paths in trees. As such, $\cap_{\lambda \in \Lambda} \Gamma(y^*, \lambda)$ must contain an immediate relative $y'$ of $y^*$ where

$$\sum_{i=1}^{|\Lambda|} \mathbf{w}_i d^2(y^*, \lambda_i) = \sum_{i=1}^{|\Lambda|} \mathbf{w}_i (d(y', \lambda_i) + 1)^2$$
$$> \sum_{i=1}^{|\Lambda|} \mathbf{w}_i d^2(y', \lambda_i).$$

Therefore, $y^* \notin \Pi(\Lambda)$ because $y^*$ is not a minimizer of $\sum_{i=1}^{|\Lambda|} \mathbf{w}_i d^2(y, \lambda_i)$. So, it must be that $y^* \in \Pi(\{\lambda_i, \lambda_j\})$ for some $\lambda_i, \lambda_j \in \Lambda$ when $y \in \Pi(\Lambda)$. $\qquad\square$

## A.4  Proof of Theorem A.6 (Algorithm B.1 correctness)

**Theorem A.6** (Algorithm B.1 correctness)**.** *Algorithm B.1 returns a locus cover for phylogenetic trees.*

*Proof.* We first prove that Algorithm B.1 will halt, then according to the stopping criterion, it is guaranteed that Algorithm B.1 returns a locus cover. Suppose the algorithm keeps running until it reaches the case in which all of the leaf nodes are included in $\Lambda$ (i.e. $\Lambda = \mathcal{Y}$), this results in the trivial locus cover of the phylogenetic tree and the algorithm halts. $\qquad\square$

## A.5  Proof of Lemma A.7 (phylogenetic tree pairwise decomposability)

**Lemma A.7** (Phylogenetic tree pairwise decomposability)**.** *Let $T = (\mathcal{V}, \mathcal{E})$ be a tree with $\mathcal{Y} = \text{Leaves}(T)$ and $\Lambda \subseteq \mathcal{Y}$. Then $\Pi(\Lambda)$ is pairwise decomposable.*

*Proof.* Suppose to reach a contradiction that $\exists y^* \in \mathcal{Y}$ where $y^* \in \Pi(\Lambda)$, but $y^* \notin \Pi(\{\lambda_i, \lambda_j\})$ $\forall \lambda_i, \lambda_j \in \Lambda$. In other words, $y^*$ is a Fréchet mean for some weighting on $\Lambda$, but $y^*$ is not a Fréchet mean for any weighting for any two vertices within $\Lambda$.

For arbitrary $\lambda_i, \lambda_j \in \Lambda$, define the closest vertex to $y^*$ on the path between $\lambda_i$ and $\lambda_j$ as

$$v'(\lambda_i, \lambda_j) = \underset{v \in \Gamma(\lambda_i, \lambda_j)}{\arg\min} \ d(y^*, v).$$

Also, let

$$\{\lambda_1, \lambda_2\} = \underset{\lambda_i, \lambda_j \in \Lambda}{\arg\min} \ d(y^*, v'(\lambda_i, \lambda_j)),$$

the pair of vertices in $\Lambda$ whose path $\Gamma(\lambda_1, \lambda_2)$ passes *closest* to $y^*$. To avoid notational clutter, define $v' = v'(\lambda_1, \lambda_2)$. Thus, $v'$ represents the closest vertex to $y^*$ lying on the path between any two $\lambda_i, \lambda_j \in \Lambda$.

Because $v'$ lies on the path $\Gamma(\lambda_1, \lambda_2)$, there exists some weighting $\mathbf{w}$ for which

$$v' = \underset{v \in \Gamma(\lambda_1, \lambda_2)}{\arg\min} \ (\mathbf{w}_1 d^2(v, \lambda_1) + \mathbf{w}_2 d^2(v, \lambda_2)).$$

For this $\mathbf{w}$, we have that $m_{\{\lambda_1, \lambda_2\}}(\mathbf{w}) = \arg\min_{y \in \mathcal{Y}} d(y, v')$, the set of leaf nodes of smallest distance from $v'$. By the initial assumption that $y^* \notin \Pi(\{\lambda_1, \lambda_2\})$, we have that $y^* \notin m_{\{\lambda_1, \lambda_2\}}(\mathbf{w})$. Let $y'$ be a vertex of $m_{\{\lambda_i, \lambda_j\}}(\mathbf{w})$ that is closest to $y^*$. Clearly, $d(y', v')^2 < d(y^*, v')^2$.

For any $v \in \mathcal{V}$, respectively define the sets of vertices (1) shared in common on the two paths $\Gamma(v, y^*)$ and $\Gamma(v, y')$, (2) on the path $\Gamma(v, y^*)$ that are not on $\Gamma(v, y')$, and (3) on the path $\Gamma(v, y')$ that are not on $\Gamma(v, y^*)$ as

$$d_C(v) = \Gamma(v, y^*) \cap \Gamma(v, y'),$$
$$d_{y^*}(v) = \Gamma(v, y^*) \setminus \Gamma(v, y'), \text{ and}$$
$$d_{y'}(v) = \Gamma(v, y') \setminus \Gamma(v, y^*).$$

We have that $|d_{y^*}(v)| + |d_{y^*}(v)| + 1 = |\Gamma(y^*, y')|$ (the difference by 1 represents the final vertex shared in common by both paths) and also that $|d_{y'}(v')| < |d_{y^*}(v')|$ since $y'$ is closer to $v'$ than $y^*$ is.

Suppose $\exists \lambda_3, \lambda_4 \in \Lambda$ yielding

$$u' = v'(\lambda_3, \lambda_4) \in \Gamma(y^*, y') \text{ such that } d^2(y^*, u') \le d^2(y', u').$$

This is to say that $u'$ is defined analogously to $v'$, with the additional restrictions that $u'$ lie on the path $\Gamma(y^*, y')$ and $u'$ be at least as close to $y^*$ as to $y'$. Then, $|d_{y^*}(u')| < |d_{y^*}(v')|$ implying $d^2(y^*, u') < d^2(y^*, v')$, which contradicts the definition of $v'$.

Suppose now that $\exists \lambda_3, \lambda_4 \in \Lambda$ yielding

$$u' = v'(\lambda_3, \lambda_4) \notin \Gamma(y^*, y') \text{ such that } d^2(y^*, u') \le d^2(y', u').$$

This is to say that $u'$ is again defined analogously to $v'$, but $u'$ is not on $\Gamma(y^*, y')$ and is at least as close to $y^*$ as to $y'$. Then, either $\cap_{\lambda \in \Lambda} d_C(\lambda) \ne \emptyset$ (i.e. all lambdas lie in the same branch off of $\Gamma(y', y^*)$), in which case the path between any two lambdas will have the same closest node $v'$ to $y^*$, or it is possible to choose a $\lambda_5 \in \Lambda$ such that $d_C(\lambda_3) \cap d_C(\lambda_5) = \emptyset$. In this situation, it must be that $\Gamma(\lambda_3, \lambda_5) \cap \Gamma(y', y^*) \ne \emptyset$, which implies there exists a vertex $t \in \Gamma(\lambda_3, \lambda_5)$ such that $d^2(y^*, t) < d^2(y^*, u')$ which violates the definition of $u'$.

Altogether, for all $v \in \Gamma(\lambda_i, \lambda_j)$ with arbitrary $\lambda_i, \lambda_j \in \Lambda$, we have that $d^2(y', v) < d^2(y^*, v)$. This implies that for all $\lambda \in \Lambda$, there exists a $y' \in \mathcal{Y}$ such that $d^2(y', \lambda) < d^2(y^*, \lambda)$ and therefore that $y^* \notin \Pi(\Lambda)$ which contradicts that $y^* \in \Pi(\Lambda)$. It must be that $\forall y^* \in \mathcal{Y}$ where $y^* \in \Pi(\Lambda)$, we have $y^* \in \Pi(\{\lambda_i, \lambda_j\}) \; \forall \lambda_i, \lambda_j \in \Lambda$. $\qquad \square$

### A.6 Proof of Theorem A.8 (minimum locus cover for grid graphs)

**Theorem A.8** (Minimum locus cover for the grid graphs). *The minimum locus cover for a grid graph is the pair of vertices in the furthest opposite corners.*

*Proof.* We would like to show two things:

1. any vertex can be resolved by construction of $\boldsymbol{w}$ using the furthest pair of two vertices (i.e. opposite corners) and

2. there does not exist a locus cover of size one for grid graphs with more than one vertex.

Let $G = (\mathcal{V}, \mathcal{E})$ be a grid graph and let $\Lambda = \{\lambda_1, \lambda_2\}$ be a set of two vertices who are on opposite corners of $G$ (i.e., perhipheral vertices achieving the diameter of $G$). For notational convenience, set $\boldsymbol{\Lambda} = (\lambda_1, \lambda_2)$ We can reach any interior vertex by an appropriate setting the weight vector $\boldsymbol{w} = [\boldsymbol{w}_1, \boldsymbol{w}_2]$. Then set $\boldsymbol{w} = \left( \frac{d(v, \lambda_2)}{d(\lambda_1, \lambda_2)}, \frac{d(v, \lambda_1)}{d(\lambda_1, \lambda_2)} \right)$. Finally, the Fréchet mean is given by

$$
\begin{aligned}
m_{\boldsymbol{\Lambda}}(\boldsymbol{w}) &= \arg\min_{y \in \mathcal{Y}} \frac{d(v, \lambda_2) d^2(y, \lambda_1) + d(v, \lambda_1) d^2(y, \lambda_2)}{d(\lambda_1, \lambda_2)} \\
&= \arg\min_{y \in \mathcal{Y}} d(v, \lambda_2) d^2(y, \lambda_1) + d(v, \lambda_1) d^2(y, \lambda_2) \\
&\ni v.
\end{aligned}
$$

Hence $\Lambda$ is a locus cover. We can clearly see that $\Lambda$ is a minimum locus cover—the only way to obtain a smaller set $\Lambda'$ is for it to include only a single vertex. However, if $\Lambda' = \{v'\}$ contains only a single vertex, it cannot be a locus cover so long as $G$ contains more than one vertex—$v'$ is always guaranteed to be a unique minimizer of the Fréchet mean under $\Lambda'$ which misses all other vertices in $G$. Thus $\Lambda$ is a minimum locus cover. $\qquad \square$

## A.7 Proof of Lemma A.9 (loci of grid subspaces)

**Lemma A.9** (Locus of grid subspaces). *Given any pair of vertices in $\Lambda$, we can find a subset $G'$ of the original grid graph $G = (\mathcal{Y}, \mathcal{E})$ which takes the given pair as two corner. $\Pi(\Lambda)$ equals to all the points inside $G'$.*

*Proof.* The result follows from application of Theorem A.8 to the choice of metric subspace. $\square$

## A.8 Proof of Lemma A.10 (grid pairwise decomposability)

**Lemma A.10** (Grid pairwise decomposability). *Let $G = (\mathcal{Y}, \mathcal{E})$ be a grid graph and $\Lambda \subseteq \mathcal{Y}$. Then $\Pi(\Lambda)$ is pairwise decomposable.*

*Proof.* Suppose we have a grid graph $G = (\mathcal{Y}, \mathcal{E})$ and a vertex $\hat{y} \in \Pi(\Lambda)$ with $\Lambda \subseteq \mathcal{Y}$. Due to the fact that $\hat{y} \in \Pi(\Lambda)$ and the fact that $G$ is a grid graph, we have that $\hat{y} \in \Gamma(\lambda_\alpha, \lambda_\beta)$ for some $\lambda_\alpha, \lambda_\beta \in \Lambda$. Then by Lemma A.10,

$$\hat{y} \in \Pi(\{\lambda_\alpha, \lambda_\beta\}) \subseteq \cup_{\lambda_i, \lambda_j \in \Lambda} \Pi(\{\lambda_i, \lambda_j\}).$$

Therefore loci on grid graphs are pairwise decomposable. $\square$

## A.9 Proof of Theorem A.11 (no nontrivial locus covers for complete graphs)

**Theorem A.11** (Trivial locus cover for the complete graph). *There is no non-trivial locus cover for the complete graph.*

*Proof.* We show that there is no nontrivial locus cover for complete graphs by showing that removing any vertex from the trivial locus cover renders that vertex unreachable. We proceed by strong induction on the number of vertices, $n$.
**Base case** We first set $n = 3$. Let $K_3 = (\mathcal{V}, \mathcal{V}^2)$ be the complete graph with three vertices: $\mathcal{V} = \{v_1, v_2, v_3\}$, and without loss of generality, let $\Lambda = \{v_2, v_3\}$ be our set of observed classes. There are two cases on the weight vector $\boldsymbol{w} = [w_2, w_3]$:
Case 1: Suppose $\boldsymbol{w} \notin \mathrm{int}\Delta^2$. This means that either $w_2 = 1$ or $w_3 = 1$—which leads to the Fréchet mean being either $v_2$ or $v_3$, respectively. Neither of these instances correspond to $v_1$ being a minimizer.
Case 2: Suppose $\boldsymbol{w} \in \mathrm{int}\Delta^2$. Then the Fréchet mean is given by

$$m_{\boldsymbol{\lambda}}(\boldsymbol{w}) = \arg\min_{y \in \mathcal{Y}} w_2 d^2(y, v_2) + w_3 d^2(y, v_3)$$

Assume for contradiction that $v_1 \in \Pi(\boldsymbol{\Lambda})$:

$$
\begin{aligned}
w_2 d^2(v_1, v_2) + w_3 d^2(v_1, v_3) &= w_2 + w_3 \\
&> w_3 &&\text{because } \boldsymbol{w} \in \mathrm{int}\Delta^2 \\
&= w_2 d^2(v_2, v_2) + w_3 d^2(v_2, v_3).
\end{aligned}
$$

Therefore, $v_1 \notin \Pi(\boldsymbol{\Lambda})$. This is a contradiction. Thus there is no nontrivial locus cover for $K_3$.

**Inductive step**: Let $K_{n-1} = (\mathcal{V}, \mathcal{V}^2)$ be the complete graph with $n - 1$ vertices. Assume that there is no nontrivial locus cover for $K_{n-1}$. We will show that there is no nontrivial locus cover for $K_n$. Let the weight vector be $\boldsymbol{w} = [w_1, ... w_{n-1}]$ corresponding to vertices $v_1, ..., v_{n-1} \in \mathcal{V}$ with $\Lambda = \{v_1, ..., v_{n-1}\}$. We want to show that $v_n \notin \Pi(\boldsymbol{\Lambda})$. We proceed by cases on $\boldsymbol{w}$.
Case 1: Suppose $\boldsymbol{w} \notin \mathrm{int}\Delta^{n-2}$ where $m$ entries of $\boldsymbol{w}$ are zero, then by strong induction we know that $v_n \notin \Pi(\boldsymbol{\Lambda}) = \{v_i\}_{i=1}^{n-m}$, i.e., there is no nontrivial locus cover for $K_{n-m}$.
Case 2: Suppose $\boldsymbol{w} \in \mathrm{int}\Delta^{n-2}$.

Assume for contradiction that $v_n \in \Pi(\Lambda)$. Using this assumption, we obtain the following

$$
\begin{aligned}
\sum_{i=1}^{n-1} w_i d^2(v_n, v_i) &= \sum_{i=1}^{n-1} w_i \\
&> \sum_{i=1}^{n-2} w_i \qquad\qquad\qquad \text{because } \boldsymbol{w} \in \mathrm{int}\Delta^{n-2} \\
&= \sum_{i=1}^{n-1} w_i d^2(v_{n-1}, v_i).
\end{aligned}
$$

Therefore, $v_n$ is not a minimizer, so $v_n \notin \Pi(\Lambda)$. This is a contradiction, hence $\boldsymbol{K_n}$ has no nontrivial locus cover. $\qquad\square$

### A.10 Proof of Theorem 4.8 (active next-class selection for trees)

*Proof.* We will prove the result by showing that the two optimization problems are equivalent. Let $v$ be a solution to the following optimization problem:

$$
\begin{aligned}
&\underset{y \in \mathcal{Y} \setminus \Pi(\Lambda)}{\arg\max} && d(y, b) \\
&\text{s.t.} && b \in \partial_{\mathrm{in}} T', \\
& && \Gamma(b, y) \setminus \{b\} \subseteq \mathcal{Y} \setminus \Pi(\Lambda),
\end{aligned}
$$

where $\partial_{\mathrm{in}} T'$ is the inner boundary of $T'$, the subgraph of $T$ whose vertices are $\Pi(\Lambda)$. This optimization problem can be equivalently rewritten as

$$
\begin{aligned}
&\underset{y \in \mathcal{Y} \setminus \Pi(\Lambda)}{\arg\max} && |\Gamma(y, b)| \\
&\text{s.t.} && b \in \partial_{\mathrm{in}} T', \\
& && \Gamma(b, y) \setminus \{b\} \subseteq \mathcal{Y} \setminus \Pi(\Lambda),
\end{aligned}
$$

and we can furthermore introduce additional terms that do not change the maximizer

$$
\begin{aligned}
&\underset{y \in \mathcal{Y} \setminus \Pi(\Lambda)}{\arg\max} && |\cup_{\lambda_i, \lambda_j \in \Lambda} \Gamma(\lambda_i, \lambda_j) \cup \Gamma(b, y)| \\
&\text{s.t.} && b \in \partial_{\mathrm{in}} T', \\
& && \Gamma(b, y) \setminus \{b\} \subseteq \mathcal{Y} \setminus \Pi(\Lambda).
\end{aligned}
$$

Equivalently, we can also connect $b$ to one of the elements of $\Lambda$

$$
\begin{aligned}
&\underset{y \in \mathcal{Y} \setminus \Pi(\Lambda)}{\arg\max} && |\cup_{\lambda_i, \lambda_j \in \Lambda} \Gamma(\lambda_i, \lambda_j) \cup \Gamma(\lambda_i, b) \cup \Gamma(b, y)| \\
&\text{s.t.} && b \in \partial_{\mathrm{in}} T', \\
& && \Gamma(b, y) \setminus \{b\} \subseteq \mathcal{Y} \setminus \Pi(\Lambda).
\end{aligned}
$$

Due to the uniqueness of paths in trees, this optimization problem also has the following equivalent form without any dependence on $b$:

$$
\begin{aligned}
v \in \underset{y \in \mathcal{Y} \setminus \Lambda}{\arg\max} |\cup_{\lambda_i, \lambda_j \in \Lambda} \Gamma(\lambda_i, \lambda_j) \cup \Gamma(\lambda_i, y)| &= \underset{y \in \mathcal{Y} \setminus \Lambda}{\arg\max} |\cup_{\lambda_i, \lambda_j \in \Lambda} \Pi(\{\lambda_i, \lambda_j\}) \cup \Pi(\{\lambda_i, y\})| \\
&= \underset{y \in \mathcal{Y} \setminus \Lambda}{\arg\max} |\cup_{\lambda_i, \lambda_j \in \Lambda \cup \{y\}} \Pi(\{\lambda_i, \lambda_j\})| \\
&= \underset{y \in \mathcal{Y} \setminus \Lambda}{\arg\max} |\Pi(\Lambda \cup \{y\})|
\end{aligned}
$$

using Lemma A.5.

Therefore $v$ is a maximizer of $|\Pi(\Lambda \cup \{v\})|$, as required. $\qquad\square$

# B Algorithms and Time Complexity Analyses

We provide time complexity analyses for Algorithms B.1, B.2, and B.3.

## B.1 Analysis of Algorithm B.1 (locus cover for phylogenetic trees)

We provide Algorithm B.1 with comments corresponding to the time complexity of each step.

---
**Algorithm 1** Locus cover for phylogenetic trees

---
**Require:** phylogenetic tree $T = (\mathcal{V}, \mathcal{E})$, $\mathcal{Y} = \text{Leaves}(T)$

$N \leftarrow |\mathcal{Y}|$

$P \leftarrow \text{sortbylength}([\Gamma(y_i, y_j)]_{i,j \in [N]})$ $\qquad \triangleright N|\mathcal{E}| + N^2 \log N$

$P \leftarrow \text{reverse}(P)$ $\qquad \triangleright O(N^2)$

$\Lambda \leftarrow \emptyset$

**for** $\Gamma(y_i, y_j)$ in $P$ **do** $\qquad \triangleright O(N^2)$

$\quad$ **if** $\Pi(\Lambda) = \mathcal{Y}$ **then** $\qquad \triangleright O(K^2 D \max\{N|\mathcal{E}|, N^2 \log N\})$

$\quad\quad$ **return** $\Lambda$

$\quad$ **else**

$\quad\quad$ $\Lambda \leftarrow \Lambda \cup \{y_i, y_j\}$

$\quad$ **end if**

**end for**

---

Combining these, we obtain the following time complexity:

$$O(N|\mathcal{E}| + N^2 \log N + N^2 + N^2 K^2 D \max\{N|\mathcal{E}|, N^2 \log N\}) = O(N^2 K^2 D \max\{N|\mathcal{E}|, N^2 \log N\}).$$

## B.2 Analysis of Algorithm B.2 (computing a pairwise decomposable locus)

We first provide Algorithm B.2 here, with comments corresponding to the time complexity of each step.

---
**Algorithm 2** Computing a pairwise decomposable locus

---
**Require:** $\Lambda$, $\mathcal{Y}$, $G = (\mathcal{V}, \mathcal{E})$

$\Pi \leftarrow \emptyset$

$D \leftarrow \text{diam}(G)$ $\qquad \triangleright O(N|\mathcal{E}| + N^2 \log N)$

**for** $\lambda_i, \lambda_j \in \Lambda$ **do** $\qquad \triangleright O(K^2)$

$\quad$ **for** $w_1$ in $\left\{\frac{0}{D}, \frac{1}{D}, ..., \frac{D}{D}\right\}$ **do** $\qquad \triangleright O(D)$

$\quad\quad$ $\mathbf{w} \leftarrow [w_1, 1 - w_1]$

$\quad\quad$ $\Pi \leftarrow \Pi \cup m_{\{\lambda_i, \lambda_j\}}(\mathbf{w})$ $\qquad \triangleright O(N|\mathcal{E}| + N^2 \log N)$

$\quad$ **end for**

**end for**

**return** $\Pi$

---

We first compute the diameter of the graph $G = (\mathcal{Y}, \mathcal{E})$ with $N = |\mathcal{Y}|$, which is done in $O(N|\mathcal{E}| + N^2 \log N)$ time using Dijkstra's algorithm to compute the shortest paths between all pairs of vertices. We then iterate over all pairs of elements in $\Lambda$ with $K = |\Lambda|$, which amounts to $O(K^2)$ iterations. Within this, we perform $O(D)$ computations of the Fréchet mean, for which each iteration requires $O(N|\mathcal{E}| + N^2 \log N)$ arithmetic operations or comparisons. Combining these, the final time complexity is

$$O(N|\mathcal{E}| + N^2 \log N + K^2 D(N|\mathcal{E}| + N^2 \log N)) = O(K^2 D \max\{N|\mathcal{E}|, N^2 \log N\}).$$

## B.3 Analysis of Algorithm B.3 (computing a generic locus)

We provide Algorithm B.3 with comments corresponding to the time complexity of each step.

Following a similar argument from the analysis of Algorithm B.2, the time complexity is

$$O(N|\mathcal{E}| + N^2 \log N + D^K(N|\mathcal{E}| + N^2 \log N)) = O(D^K),$$

**Algorithm 3** Computing a generic locus

---

**Require:** $\Lambda, \mathcal{Y}, G = (\mathcal{V}, \mathcal{E})$
  $\Pi \leftarrow \emptyset$
  $D \leftarrow \text{diam}(G)$                                                            $\triangleright O(N|\mathcal{E}| + N^2 \log N)$
  **for** $\mathbf{w}$ in $\left\{\frac{0}{D}, \frac{1}{D}, ..., \frac{D}{D}\right\}^{|\Lambda|}$ **do**                         $\triangleright O(D^K)$
    $\Pi \leftarrow \Pi \cup m_{\Lambda}(w)$                                  $\triangleright O(N|\mathcal{E}| + N^2 \log N)$
  **end for**
  **return** $\Pi$

---

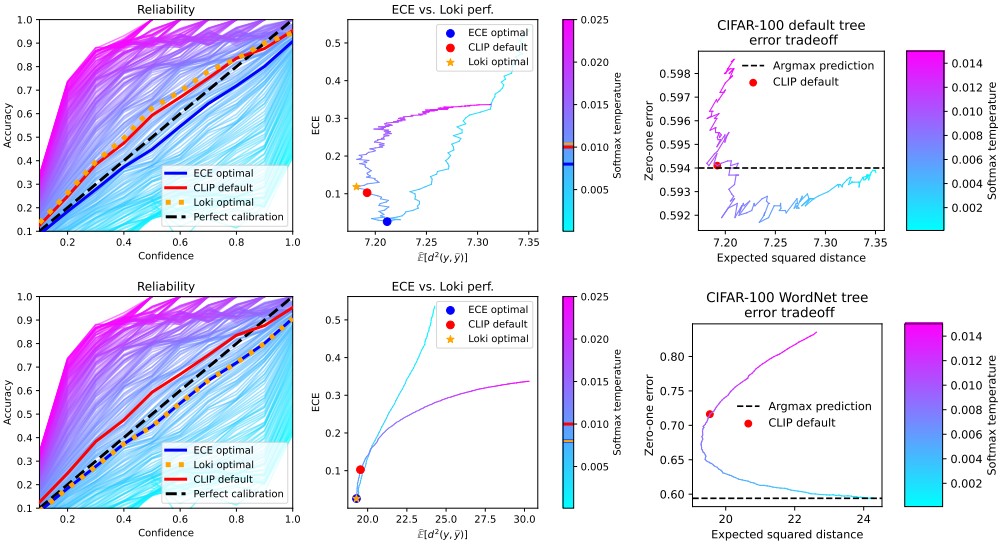

Figure 4: **CLIP on CIFAR-100 with the WordNet hierarchy.** (Left) Reliability diagrams across a range of Softmax temperatures, highlighting the CLIP default temperature, the optimal temperature for LOKI, and the minimizer of the Expected Calibration Error (ECE). All three are well-calibrated. (Center) Tradeoff between optimizing for ECE and the expected squared distance. As with the reliability diagrams, the CLIP default temperature, the LOKI-optimal temperature, and the ECE-optimal temperature are similar. (Right) Tradeoff between optimizing for zero-one error and the expected squared distance. Depending on the metric space, temperature scaling can improve *accuracy*.

where $K$ is the number of observed classes.

## C   Additional Experiments

### C.1   Additional calibration experiments

In the main text, we evaluated the effect of calibrating the Softmax outputs on the performance of LOKI using a metric space based on the internal features of the CLIP model. Here, we perform the same analysis using two external metric spaces.

**Setup**   We perform our calibration analysis using the default and WordNet tree.

**Results**   Our results are shown in Figure 4. Like our experiment using an internally-derived metric, we find that the optimal Softmax temperature is close to the CLIP default and the optimal temperature for calibration. For the default tree, we again found that temperature scaling can be used to improve accuracy using LOKI.

**Setup**   We calibrate via Softmax temperature scaling [13] using CLIP on CIFAR-100. We do not use an external metric space, and instead use Euclidean distance applied to the CLIP text encoder.

**Results** The reliability diagram in Figure 3 shows that the optimal Softmax temperature for LOKI is both close to the default temperature used by CLIP and to the optimally-calibrated temperature. In Figure 3 (right), we find that appropriate tuning of the temperature parameter *can lead to improved accuracy with CLIP*, even when no external metric space is available.

## C.2  Ablation of LOKI formulation

LOKI is based on the Fréchet mean, which is defined as $\arg\min_{y \in \mathcal{Y}} \sum_{i=1}^{K} \mathbf{P}_{\lambda_i|x} d^2(y, \lambda_i)$. However, this is not the only approach that can be considered. For example, the Fréchet *median*, often used in robust statistics, is defined as $\arg\min_{y \in \mathcal{Y}} \sum_{i=1}^{K} \mathbf{P}_{\lambda_i|x} d(y, \lambda_i)$, without squaring the distances. More generally, we define $\arg\min_{y \in \mathcal{Y}} \sum_{i=1}^{K} \mathbf{P}_{\lambda_i|x} d^\beta(y, \lambda_i)$ and evaluate different choices of $\beta$.

**Setup** We conduct this experiment on ImageNet using SimCLR as our pretrained classifier with $K = 250$, 500, and 750 randomly selected classes.

**Results** From this analysis, we conclude that using the Fréchet mean is the optimal formulation for Loki, as it achieved the lowest mean squared distance for all settings of $K$.

Table 4: Expected squared distances of SimCLR+Loki alternatives on ImageNet. We ablate over the choice of distance exponent in LOKI (where $\beta = 2$, corresponding to the Fréchet mean), including the Fréchet median ($\beta = 1$). That is, we tune $\beta : \hat{y} \in \arg\min_{y \in \mathcal{Y}} \sum_{i=1}^{K} \mathbf{P}_{\lambda_i|x} d^\beta(y, \lambda_i)$ and find that the optimal setting is $\beta = 2$, corresponding to LOKI.

| $\beta =$ | $K = 250$ | $K = 500$ | $K = 750$ |
|---|---|---|---|
| 0.5 | 61.28 | 46.57 | 36.31 |
| 1 | 52.98 | 41.06 | 33.14 |
| **2 (Loki)** | **46.78** | **37.76** | **29.88** |
| 4 | 47.99 | 39.58 | 34.65 |
| 8 | 65.29 | 58.42 | 54.90 |

## C.3  Comparison across metric spaces

Expected squared distances cannot be directly compared across metric spaces, as they may be on different scales. Our solution is to use a form of normalization: we divide the expected squared distance by the square of the graph diameter. This brings all of the values to the 0-1 range, and since $\mathbb{E}[d^2(y, \hat{y})]/\text{diam}(G)^2$ indeed also generalizes the 0-1 error, this enables comparison between 0-1 errors and those from different metric spaces. We provide these results in Table 1, again for our CLIP experiments on CIFAR-100. This evaluation metric enables us to determine which metric spaces have geometry best 'fit' to our pretrained models.

**Setup** Using $\mathbb{E}[d^2(y, \hat{y})]/\text{diam}(G)^2$, we re-evaluate our CLIP results on CIFAR-100 shown in Table 1 for the ResNet-50 architecture.

**Results** For CIFAR-100, we observe that the WordNet metric space resulted in the lowest error and therefore has the best geometry.

Table 5: Comparison across metric spaces for CLIP on CIFAR-100 by normalizing by the squared diameter of the metric space: $\mathbb{E}[d^2(y, \hat{y})]/\text{diam}(G)^2$.

| Metric space $G$ | $\text{diam}(G)^2$ | $\mathbb{E}[d^2(y, \hat{y})]/\text{diam}(G)^2$ |
|---|---|---|
| Complete graph | 1 | 0.5941 (0-1 error) |
| Default tree | 16 | 0.4493 |
| **WordNet tree** | 169 | **0.1157** |
| CLIP features | 0.9726 | 0.2686 |

# D    Experimental Details

In this section, we provide additional details about our experimental setup.[3]

## D.1    CLIP experiments

All experiments are carried out using CLIP frozen weights. There are no training and hyperparameters involved in experiments involving CLIP, except for the Softmax temperature in the calibration analysis. We eveluted using the default CIFAR-100 test set. The label prompt we use is "a photo of a [class_name]."

## D.2    ImageNet experiments

To construct our datasets, we randomly sample 50 images for each class from ImageNet as our training dataset then use the validation dataset in ImageNet to evaluate LOKI's performance. We extract the image embeddings by using SimCLRv1[4] and train a baseline one-vs-rest classifier. We use WordNet phylogenetic tree as the metric space. The structure of phylogenetic tree can be found at here[5]. We test different numbers of observed classes, $K$, from 1000 classes. Observed classes are sampled in two ways, uniformly and Gibbs distribution with the concentration parameter 0.5.

## D.3    LSHTC experiments

We generate a summary graph of the LSHTC class graph (resulting in supernodes representing many classes) by iteratively:

1. randomly selecting a node or supernode
2. merging its neighbors into the node to create a supernode

until the graph contains at most 10,000 supernodes. In the LSHTC dataset, each datapoint is assigned to multiple classes. We push each class to its supernode, then apply majority vote to determine the final class. We test different numbers of observed classes, $K$, from 10,000 classes. We collect datapoints which are in the observed classes. Then split half of dataset as training dataset and make the rest as testing dataset, including those datapoints which are not in the observed classes. We train a baseline classifier using a 5-NN model and compare its performance with LOKI.

# E    Broader Impacts and Limitations

As a simple adaptor of existing classifiers, it is possible for our method to inherit biases and failure modes that might be present in existing pretrained models. On the other hand, if the possibility of harmful mis-classifications are known a priori, information to mitigate these harms can be baked into the metric space used by LOKI. Finally, while we have found that LOKI often works well in practice, it is possible for the per-class probabilities that are output by a pretrained model to be sufficiently mis-specified relative to the metric over the label space, or for the metric itself to be mis-specified. Nonetheless, we have found that off-the-shelf or self-derived metric spaces to work well in practice.

# F    Random Locus Visualizations

In Figures 1, 5, 6, 7, 8, and 9, we provide visualizations of classification regions of the probability simplex when using LOKI with only three observed classes out of 100 total classes, and different types of random metric spaces. The first example in Figure 1 shows the prediction regions on the probability simplex when using standard $\arg \max$ prediction—the three regions correspond to predicting one of the three classes (0, 39, and 99) and no regions corresponding to any of the other classes $\{1, ..., 38, 40, ..., 98\}$. We compute these regions using a version of Algorithm B.3, and while

---

[3]Code implementing all of our experiments is available here: `https://github.com/SprocketLab/loki`.
[4]Embeddings are extracted from the checkpoints stored at: `https://github.com/tonylins/simclr-converter`
[5]`https://github.com/cvjena/semantic-embeddings/tree/master/ILSVRC`

it does have exponential time complexity, the exponent is only three in this case since we consider only three observed classes.

On the other hand, the other three examples in Figure 1 and Figures 5, 6, 7, 8, and 9 all show the prediction regions on the probability simplex when using LOKI. Figure 5 shows this for random tree graphs. Here, the prediction regions are striped or contain a single triangle-shaped region in the center—these correspond, respectively, to intermediate classes along branches of the tree leading up from the observed class and the prediction region formed by the common-most parent node. Figure 6 shows similar regions, although these are more complex and are thus more difficult to interpret, as phylogenetic trees are metric subspaces equipped with the induced metric from trees. Furthermore, in order to generate phylogenetic trees with 100 leaves, we needed to create much larger trees than the ones used for Figure 5, which led to narrower prediction regions due to the higher graph diameter. Finally, Figures 7, 8, and 9 each show the prediction regions using LOKI when the metric space is a random graph produced using Watts-Strogatz, Erdős-Rényi, and Barabási–Albert, models respectively. These prediction regions are more complex and represent complex relationships between the classes.

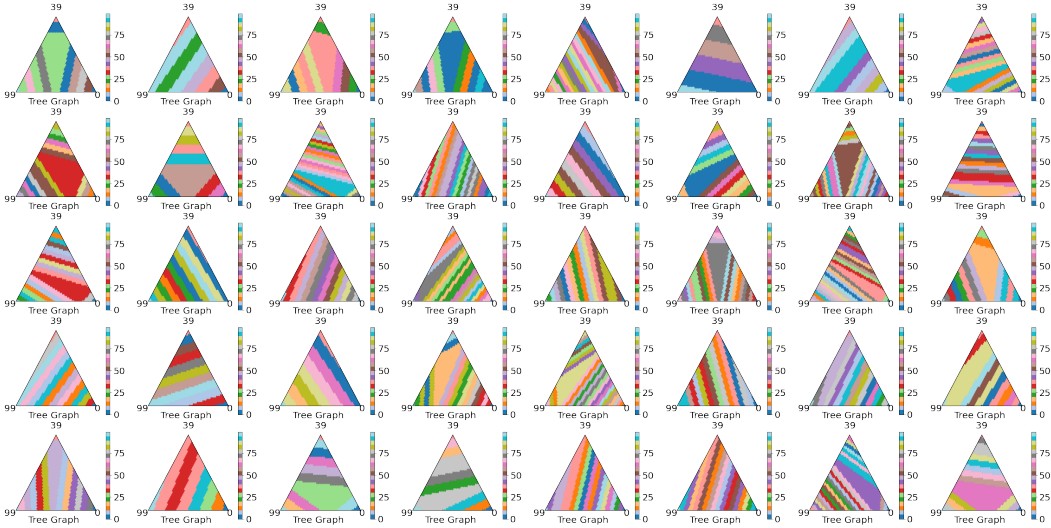

Figure 5: Classification regions in the probability simplex of 3-class classifiers faced with a 100-class problem where the classes are related by random trees graphs.

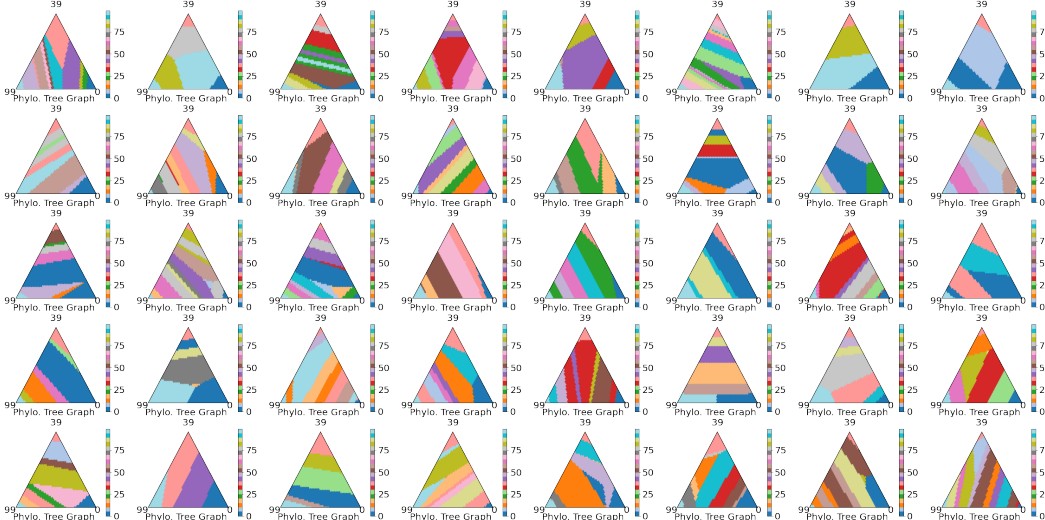

Figure 6: Classification regions in the probability simplex of 3-class classifiers faced with a 100-class problem where the classes are related by random phylogenetic trees graphs.

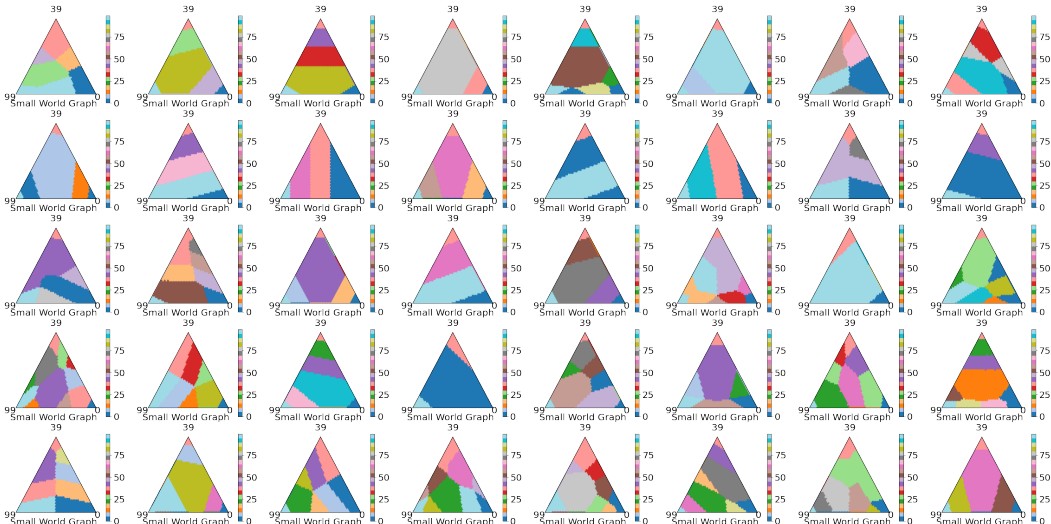

Figure 7: Classification regions in the probability simplex of 3-class classifiers faced with a 100-class problem where the classes are related by random small-world graphs generated by a Watts–Strogatz model.

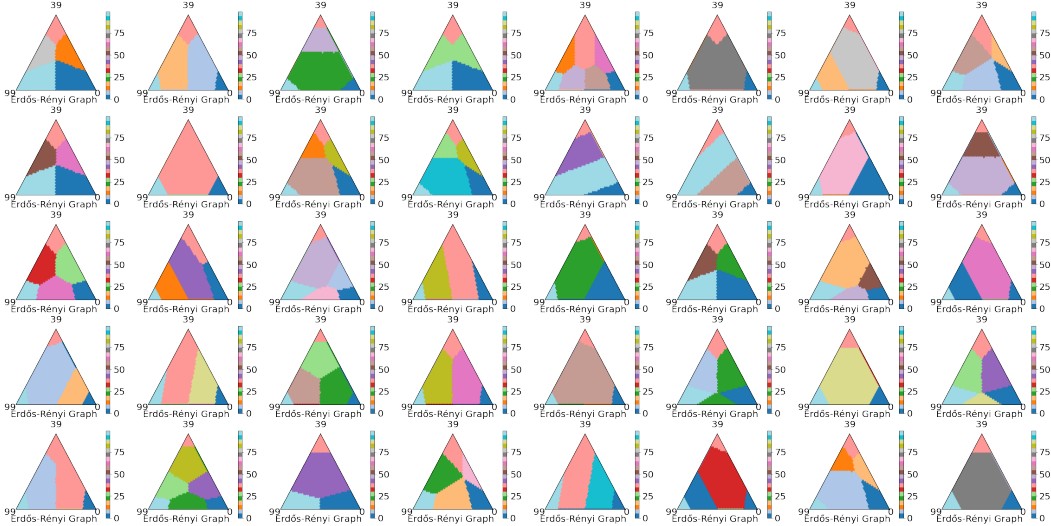

Figure 8: Classification regions in the probability simplex of 3-class classifiers faced with a 100-class problem where the classes are related by random Erdős-Rényi graphs.

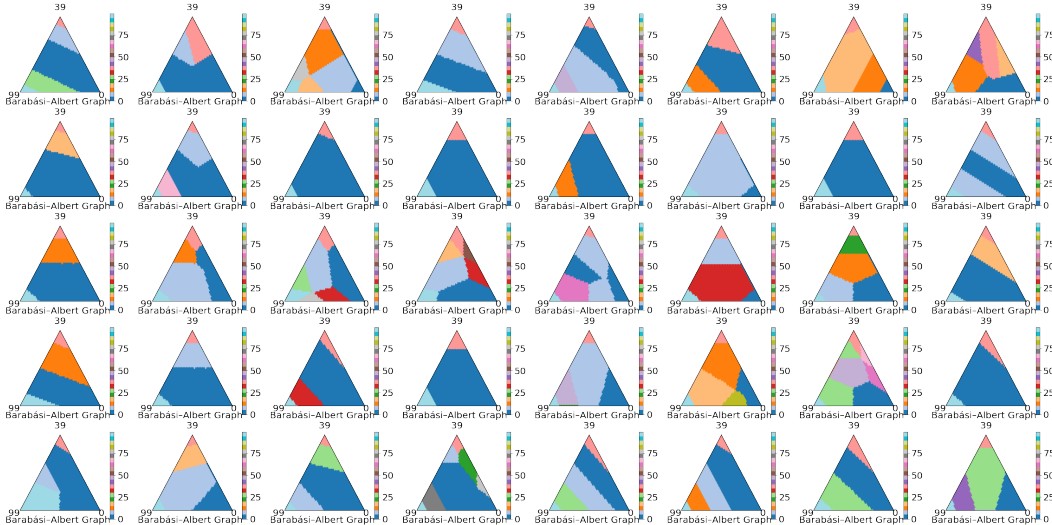

Figure 9: Classification regions in the probability simplex of 3-class classifiers faced with a 100-class problem where the classes are related by random Barabási–Albert graphs.

