# OpenReview forum: "Geometry-Aware Adaptation for Pretrained Models"
_NeurIPS.cc/2023/Conference — NeurIPS 2023 poster_

### Official Review · Reviewer_dZLD · 2023-07-02

**Soundness:** 3 good
**Presentation:** 3 good
**Contribution:** 2 fair
**Rating:** 5
**Confidence:** 3

**Summary:**

In this paper, the authors explore the adaptation of pretrained models from a geometric perspective. Specifically, this paper leverages the label information as a metric space and proposes a simple approach to predict new classes based on the pretrained models’ zero-shot prediction without any further training. This paper also provides some theoretical analysis w.r.t learning-theoretical results, locus cover and active next-class selection. In the empirical results, this paper conducts experiments on several datasets and verify the effectiveness of the proposed method.

**Strengths:**



- The paper is generally well-written and easy to follow.
- The topic of adaptation on the pretrained models from the geometric perspective is interesting.
- The proposed method is simple and efficient.
- The main claims are generally supported by theoretical results.





**Weaknesses:**

- The proposed method may require some additional information such as label relationship, which may not be applicable in some real-world situations.
- It lacks theoretical analysis about the superiority of the proposed LOKI.
- The empirical results are not adequate to support the main claims.

**Questions:**

- As discussed in the Weaknesses, it is not clear whether the proposed LOKI is the optimal geometric adaptation method for enabling the prediction of unobserved classes. Are there any alternatives for Fréchet mean estimator to yielding more reliable predictions? (An example) It would benefit a lot if some empirical or theoretical comparison are added to demonstrate the effectiveness of the proposed LOKI.
- I appreciate this paper that most claims are supported by theoretical results. It would further improve the paper if more empirical results were added to support these claims. For example, some toy experiments can be provided to verify Theorem 4.8.
- In Table 1, the effectiveness of LOKI is explored on CIFAR-100. It would be more convincing to conduct experiments on more real-world datasets.
- The experiments in Table 1 seems not comprehensive. From the results, we can see that LOKI presents limited improvements on ViT-L-14 backbone. It would be better to add more discussions and more comprehensive empirical results.


**Limitations:**

Yes.

---

> ### Author Rebuttal · Authors · 2023-08-09
>
> Thank you for your review! We have responded to your points below.
>
> **On the usage of the Fréchet mean estimator**
>
> We made the choice to use the Fréchet mean because we have information about the relationships between classes. To ground our rationale in a different type of problem, consider a regression task: the output is a **scalar,** and it is possible to compare labels (e.g., one is larger than another, etc.). Labels can also be averaged, and indeed, the Bayes optimal linear predictor is just such an average—a mean.
>
> However, in our setting, we are in a far more general case than continuous spaces like $\mathbb{R}$. In fact the only information we have access to is relationships between classes, which can be expressed as distances. So we need an analogue to the mean that operates using only distances, and the canonical analogue is the Fréchet mean. Indeed, it has beautiful theoretical [1] and optimization [2] properties. Other variations of the weighted Fréchet mean have been used in supervised learning in structured prediction settings [3], although since our problem does not permit training, we require a novel approach.
>
> However, it is not the only possible choice. For example, taking an exponent of 1 instead of 2 on the distance term produces the Fréchet median, which is known to have excellent robustness properties, but lacks some of the theoretical properties we rely on. Our framework is flexible—we could obtain a new version of Loki that uses such an object as well. **We include new experimental results that evaluate the effect of changing the exponent**; please see Table 2 in our new results.
>
>
> **On experiments to verify Theorem 4.8**
>
> We agree and indeed our submission already includes exactly this! Figure 2 (right) is a synthetic experiment that shows how Theorem 4.8 can be used to optimally increase the size of the locus for tree graphs. In addition, since submission, we have included a new experiment that verifies Theorem 4.8 on ImageNet – a much more realistic setting.
>
> **On CIFAR-100 and more real world datasets**
>
> Our submission includes three real-world datasets in addition to CIFAR-100: ImageNet, PubMed, and LSHTC. While ImageNet (ILSVRC 2012) includes 1,000 classes, PubMed includes nearly 10,000, and LSHTC includes over 325,000 classes.
>
> **On limited improvements using the ViT-L-14 backbone**
>
> We were excited to find that there are substantial gains to be had by applying Loki to a variety of pretrained models from different settings: supervised models, self-supervised models, and zero-shot models including CLIP and ALIGN. We hypothesize that even greater gains can be obtained for the ViT-L-14 CLIP backbone if we were to use a more refined approach for extracting the metric space or tune the softmax temperature.
>
> [1] https://arxiv.org/abs/1609.03045
>
> [2] https://arxiv.org/abs/2003.00335
>
> [3] https://arxiv.org/abs/1605.07588

---

> > ### Comment · Reviewer_dZLD · 2023-08-16
> > **Thanks for the rebuttal**
> >
> > Dear authors,
> >
> > Thanks for the detailed replies. After reading the rebuttal and other reviews, my concerns have been adequately addressed. I would like to raise the score to '5'.
> >
> > Best,
> >
> > Reviewer dZLD

---

> > > ### Author Response · Authors · 2023-08-18
> > >
> > > We are excited to include these new results in the final version! Thank you for raising these points in your review, and we appreciate your engagement with our rebuttal! Please let us know if there are any questions we can answer about our new results or anything else about the paper. Thanks!

---

### Official Review · Reviewer_2m87 · 2023-07-06

**Soundness:** 3 good
**Presentation:** 4 excellent
**Contribution:** 3 good
**Rating:** 7
**Confidence:** 3

**Summary:**

This paper explores the concept of geometry-aware adaptation for label spaces. The paper introduces a method called "LOKI" that allows pretrained models to make predictions for classes that were not observed during training. LOKI utilizes metric space information to adapt the model's predictions to unobserved classes.

The paper discusses the theoretical foundations of LOKI, including the definition of loci and identifying locus covers in graphs. It presents algorithms for efficiently computing the locus and describes an active learning-based strategy to select the next class for observation in order to maximize the size of the locus.

Experimental results demonstrate that LOKI improves the performance of zero-shot models even without external metric space information. It adapts to label spaces with a large number of unobserved classes and outperforms baseline models in terms of mean squared distances. The paper also validates the effectiveness of the active next class selection approach, showing that it leads to larger loci compared to random selection.








**Strengths:**

1. Originality: The paper exhibits a high degree of originality in several aspects. Firstly, it introduces the concept of geometry-aware adaptation for label spaces, which is a novel approach to address the challenge of predicting unobserved classes in pretrained models. The formulation of LOKl and its integration with metric space information is a unique contribution. Additionally, the paper presents novel definitions, algorithms, and strategies related to loci, identifying locus covers, and active learning-based class selection. These original contributions set the paper apart and make it a valuable addition to the field.

2. Clarity: The paper excels in terms of clarity, making the research accessible to a wide range of readers. The authors provide clear explanations of concepts, definitions, and algorithms, ensuring that readers can understand the technical aspects of LOKI. The paper is well-structured, with logical flow and section organization.

3. Quality: The paper demonstrates a high level of quality in terms of its theoretical foundations, experimental methodology, and presentation of results. The authors provide a thorough and rigorous analysis of the problem, offering formal definitions and proving relevant theorems.

4. Significance: The paper holds significant importance in the field of machine learning and predictive modeling. It addresses a challenging problem of predicting unobserved classes in pretrained models, which has practical implications in various real-world scenarios. The introduction of LOKI and its ability to adapt predictions using metric space information expands the capabilities of pretrained models, enabling them to make more accurate predictions for classes that were not observed during training. The experimental results demonstrate the effectiveness of LOKI and its superiority over baseline models, further highlighting its significance in advancing the field.



**Weaknesses:**

1. Lack of Comparative Analysis: While the paper presents LOKI as a novel approach, it would benefit from a more comprehensive comparative analysis against existing methods or related works. Providing such a comparison would strengthen the paper's argument for the originality and effectiveness of LOKI.

2. Scalability Analysis: The paper briefly mentions the computational complexity of LOKI, but a more detailed analysis of its scalability would be valuable.

**Questions:**

1. While the paper briefly mentions the computational complexity of LOKI in term of label spaces, will time consumption of this method be also related with higher-dimensional metric spaces? When compared with original method without LOKI, how much is extra consumption of memory and runtime with LOKI?

2. When multiple external metrics are provided, can we decide which metric is most suitable in advance?

3. Is the property pairwise decomposable necessary for the improvement of LOKI? Does it only work on the dataset satisfying this property?

**Limitations:**

No potential negative societal impact of their work. Suggestions are provided in the weakness and questions.

---

> ### Author Rebuttal · Authors · 2023-08-09
>
> Thank you for your thoughtful review and for **noting the originality, clarity, quality, and significance of our work!**
>
> **On comparative analyses**
>
> We are unaware of other approaches that operate in our setting: adapting a pretrained classifier to enable the navigation of metric spaces without any additional training. **However, we can evaluate alternatives to Loki** – Loki replaces the $\text{arg max}$ with the Fréchet mean, such as the Fréchet median. We compared four alternatives to Loki in our new experimental results (see Table 2), and we found that Loki was optimal for minimizing the expected squared distance.
>
> **On scalability analyses and runtime**
>
> We demonstrate that Loki can be scaled up to the ~325,000-class LSHTC dataset by using approximations such as graph summarization. We will provide a more thorough ablation of the scalability of Loki in the final version of the paper.
>
>
> **On choosing the best metric**
>
> This is an important point! Typically, the answer would be to evaluate different metrics on a held-out validation set, but in this setting, the expected squared distance itself depends on the metric space. In order to make comparisons between metric spaces, we need the errors to be on the same scale. We provide new results that normalize all of the expected squared distances by the squared diameter of the metric space, bringing all of the errors onto the same 0-1 scale, and allows us to compare with other performance measures such as the 0-1 error. This enables tuning the choice of metric space as a hyperparameter – in the case of CLIP applied to CIFAR-100, we find that the WordNet tree is the optimal metric space of the ones that we evaluated.
>
> **On whether or not pairwise decomposability is necessary for Loki**
>
> Pairwise decomposability is only needed for efficient computation of the active data selection strategy. Loki itself does not depend on this property – Loki can be applied using a linear transformation of the softmax probabilities regardless of the properties of the metric space or loci.

---

> > ### Comment · Area_Chair_339S · 2023-08-18
> > **Reaction to authors' response**
> >
> > Dear reviewer 2m87. Has the authors' response answered your questions? Is there any other clarification you would want to request from the authors before the discussion period ends?
> >
> > AC

---

### Official Review · Reviewer_4Uud · 2023-07-07

**Soundness:** 3 good
**Presentation:** 3 good
**Contribution:** 3 good
**Rating:** 7
**Confidence:** 3

**Summary:**

The authors consider the problem of predicting examples from unseen but "known" classes. Taking inspiration from structured prediction, the paper intends to exploit the knowledge of structure in the full label space. The authors propose an alternative to the popular "argmax-over-logits" prediction, called Loki, by computing the Frechet mean with an appropriate distance metric in label space. A theoretical result on a simple logistic data model shows the sample complexity bound with Loki. Also considered is the important question of what training labels are necessary and sufficient to be able to span the whole label space during prediction. On the experimental effectiveness, the authors demonstrate reasonable percentage point improvement over baseline "argmax" standard prediction.

**Strengths:**

Strengths:
1. My knowledge in structured prediction is limited. However, it seems to me that the results in this paper are significant and interesting
2. I found the method Loki to be a neat approach to predicting unseen classes, when the distance metric for th label space is known. It is a scalable method since the computation involved is a fixed linear transformation
3. The authors have stated and and given proofs for the minimal locus cover for two non-trivial label space structures with clear motivation - phylogenetic trees and grid graphs
4. An algorithm of a "greedy" nature through an active-learning framework is then given, which is polynomial time.
5. Experimental results: it seems that Loki achieves strong results for CLIP
6. While I have not checked the proofs in detail, the methodology appears sound and rigorous.


**Weaknesses:**

Weaknesses:
1. It is not clear to me if there can be guarantees provided when the true label space metric is not available or only approximately known
2. Theorem 1 is for the case of logistic regression. It is not clear whether this is a powerful enough model for practice.
3. I have either missed this part or there needs to be a detailed discussion of the implications of Theorem 1
4. I suggest providing "error bars" or significance level for the numbers reported in Table 1
5. More experimental details would be helpful for reproducing the results
6. It is unclear to me if there are other competitive structured prediction techniques applied to pretrained models and what gains Loki makes over them, if any.


**Questions:**

Questions:
1. Line 210: "In cases involving an extremely large number of classes, it is desirable to use LOKI on the smallest 211 possible set of observed classes" - is this because sample complexity of in Theorem 1 scales as K^2? Are there other considerations?
2. I would prefer to get more explanation for the proposed use-case of grid graphs. While phylogenetic trees is more intuitive, grid graphs might benefit from a simple examples
3. Does a counterpart of Theorem 4.8 hold for grid graphs?
4. Sec 5.2: Is one-vs-rest the most competitive baseline?
5. For the result on computing loci for partially oserved label spaces, are PubMed and LSHTC the standard datasets to consider? Please provide any reasoning behind this choice. Any references will be useful
6. Fig 1 (and related ones in Appendix)- it is not immediately clear what the authors intend to demonstrate here.
7. I am curious about the impact of Loki on calibration.

**Limitations:**

Limitations:
I did not see an explicit discussion on the limitations and scope for improvement

---

> ### Author Rebuttal · Authors · 2023-08-09
>
> Thank you for the thoughtful review and for **noting the significance, theoretical contributions, and scalability of our work!**
>
> **On the true label space metric vs. approximations**
>
> Excellent question! We find that there is not a single “correct” metric space for a given problem, and that many approximations can be made without significantly impacting performance (e.g. deriving a metric from pretrained class embeddings, graph summarization, or minimum spanning tree approximations). We hypothesize that this phenomenon is due to the presence of redundancy within many of the objects commonly used in ML (representations, graph structures, etc.). This redundancy enables a variety of potentially distinct metric spaces to contain enough geometric information to be successfully used by Loki.
>
>
>
>
>
> **On logistic regression in practice**
>
> Note that this logistic regression result (LR) holds in practice for a variant of our SimCLR experiments, which involves applying it to self-supervised feature embeddings. While simple, this result applies to the broad range of modern self-supervised learning techniques that use logistic regression as part of the learning pipeline. We believe that the result can also be extended beyond LR to richer distributions and model classes, but the machinery involved in such an analysis becomes more complex **without shedding additional light on the fundamental tradeoffs.** For example, we could perform a similar analysis for two-layer neural networks, but existing results, which must serve as a component in our result, are often unwieldy and only obscure the roles played by key quantities (e.g., the graph diameter).
>
> **On implications of Theorem 1**
>
> Thank you for bringing this up – we added a detailed discussion of Theorem 1 in the updated draft. In short, we can bound individual errors in expected squared distance as a function of the squared diameter of the graph, the number of observed classes, the dimension of the features used to train the logistic classifier, the number of samples, and a problem-dependent constant. The squared errors relate to the squared diameter of the graph, as this is the maximum value that they can take on. The bound improves as we obtain more samples—as we would expect. Furthermore, since the bound has a quadratic dependence on K, we can also improve the bound if our classifier is trained on fewer classes.
>
> **On error bars and significance of Table 1**
>
> The results of Table 1 are deterministic, as we cannot retrain CLIP over multiple seeds, although we will provide error bars for our other experiments in the final version.
>
> **On more experimental details and reproducibility**
>
> We absolutely agree! We will include full experimental details and release code reproducing the results in the final version.
>
> **On other competitive structured prediction methods using pretrained models**
>
> We are unaware of other structured prediction approaches that operate in our setting: adapting a pretrained classifier to enable the navigation of metric spaces without any additional training. **However, we evaluated alternatives to Loki**, such as the Fréchet median (see common response).
>
> **On applying Loki to the smallest number of observed classes.**
>
> The reason for applying Loki to a small set of classes is twofold – practical usage (i.e., not needing to train a classifier on the entire, potentially large label space), and theoretically as a property of our bound, as you correctly pointed out. Consider the example of 2D grid graphs – in order to predict an arbitrarily large grid, we only need the four corners to form an identifying locus cover. In other words, we can predict an arbitrarily large number of classes using only a four-class classifier.
>
>
> **On real-world use-case of grid graphs**
>
> Problems in which classes represent points in space can be represented as grid graphs, e.g., predicting locations on a map, predicting the next move in a board game such as chess, and predicting atoms in a lattice where edges represent physical interactions [1].
>
> **On Theorem 4.8 for grid graphs**
>
> For bounded 2D rectangular grids, the result is trivial – over a constant four rounds of active selection, select the four corners. This leads to an identifying locus cover. If we relax the requirement that each node can be predicted uniquely, then this can be done in only two rounds of active selection by selecting opposite corners. This forms a minimal locus cover. We will include this example in the final version of the text.
>
> **On the one vs. rest baseline**
>
> We used one vs. rest logistic regression for its simplicity, although a similar result can be obtained using multiclass logistic regression. Applying logistic regression to self-supervised embeddings is typical for self-supervised learning approaches.
>
> **On reasons to compute loci on partially observed label spaces such as PubMed and LSHTC**
>
> We used these datasets because they 1. Have a large number of classes and 2. Either have a native metric space structure (LSHTC) or have class names where a metric space can be derived easily using pretrained class embeddings (PubMed). To our knowledge, there is no prior work on computing loci of label spaces in the machine learning literature from which we could have obtained datasets.
>
> **On the purpose of Figure 1**
>
> Fig. 1 visualizes the locus of 100-node graphs when applying Loki to a 3-class classifier. For example, the leftmost simplex represents prediction using the argmax – in this case, there are three equally-sized regions corresponding to the Softmax outputs of the 3-class classifier. We will clarify this in the final version.
>
> **On calibration**
>
> This is an excellent point! We have included a new analysis of the effect of calibration, as described in the common response.
>
> [1] https://arxiv.org/abs/2010.09990
>
> [2] https://arxiv.org/abs/1706.04599

---

> > ### Comment · Reviewer_4Uud · 2023-08-16
> > **Thank you for the clarifications and additional supporting results**
> >
> > I would like to thank the authors for their detailed and useful response. The calibration results are interesting and would be a good addition to the paper, providing useful guidance on tuning the temperature hyperparameter. I do not have further questions at this time and would like to keep the score of "accept".

---

> > > ### Author Response · Authors · 2023-08-18
> > >
> > > Thank you for engaging with our rebuttal, and for your positive feedback! We are excited to include the temperature scaling results in the final version of the paper as we believe that your suggestion to study calibration revealed an exciting aspect of our work. Please let us know if any additional questions come up about our new results or anything else about the paper. Thanks!

---

### Official Review · Reviewer_cqEn · 2023-07-07

**Soundness:** 3 good
**Presentation:** 3 good
**Contribution:** 3 good
**Rating:** 6
**Confidence:** 2

**Summary:**

This paper considers the zero-shot model adaptation to testing tasks that include classes not seen during training. The authors propose a post-hoc method called LOKI, which applies a class-graph-based transformation to make predictions on these unseen classes. The authors also provide the theoretical analysis and experimental results to support the proposal.

**Strengths:**

The problem considered in this paper is significant in the machine learning community, and the proposed solution is technologically reasonable.
The theoretical results are interesting and provide a solid justification for the proposed solution.
The paper is well-written and easy to follow.

**Weaknesses:**

The graph of LOKI seems heavily rely on the prior knowledge of the classes.
In the Sections 5.1 and 5.2 of this paper, experiments were conducted on CIFAR100 and ImageNet, respectively. Are there any additional results on other datasets?

**Questions:**

How can a reliable graph be obtained in real-world applications? The experiments implement it using hierarchical trees and WordNet. Does this limit the practical application of the proposal?


---
Thanks for the detailed clarifications, which have addressed my concerns. I would like to keep my score.

**Limitations:**

The authors do not provide a discussion about the limitations and potential negative societal impact.

---

> ### Author Rebuttal · Authors · 2023-08-09
>
> Thank you for the thoughtful review and for noting the significance of the problem considered in our work! We have clarified points about our experimental results and the metric spaces that we use below.
>
> **On prior knowledge of the classes**
>
> Metrics relating the classes are often readily available, but **even when they are not, our method still works**. Our experimental results in Section 5 include two such examples:
> - In Table 1, the “Internal” metric space refers to using class embeddings from the pretrained CLIP model itself, and
> - In Table 2, the metric space for our PubMed results is derived from SimCSE embeddings.
>
> In both of these cases, Loki improves over the baseline, which confirms that prior knowledge of the classes is not required.
>
> **On experimental results beyond CIFAR-100 and ImageNet**
>
> Yes, our submission contains experiments on PubMed and the LSHTC datasets, please see Tables 2 and 3 in our submission. Both **datasets contain substantially more classes than CIFAR-100 and ImageNet (ILSVRC 2012)**, with PubMed containing almost 10,000 classes and LSHTC containing 325,000 classes. We chose these datasets in order to have diversity both in terms of the data domain and the size and structure of the metric space.
>
> **On obtaining metric spaces in the real world**
>
> Great question! Our experiments include graphs from a variety of sources that are **not limited to hierarchies.** For example, our PubMed and CIFAR-100 experiments include metrics that were derived from models where we naively applied the Euclidean distance to off-the-shelf embeddings and internal representations of the class names. The resulting metric spaces are not hierarchical. Obtaining these graphs simply required applying a standard procedure rather than needing any specialized domain knowledge. This procedure can be followed in any setting where embeddings are available.
>
> A natural question to ask is whether different approaches to obtaining graphs lead to similar results. Excitingly, in our CIFAR-100 experiments, **we show that Loki can lead to improved performance across three different metric spaces for the same problem,** suggesting that Loki is robust to changes in the metric space.

---

> > ### Comment · Reviewer_cqEn · 2023-08-13
> >
> > Thank you for your detailed clarifications, which have addressed my concerns. I will keep my score and recommend acceptance of this paper.

---

> > > ### Author Response · Authors · 2023-08-18
> > >
> > > Thank you for engaging with our rebuttal, and for your positive feedback! Please let us know if there are any questions we can answer about our new results or anything else about the paper. Thanks!

---

### Official Review · Reviewer_zgY7 · 2023-07-27

**Soundness:** 3 good
**Presentation:** 3 good
**Contribution:** 2 fair
**Rating:** 5
**Confidence:** 4

**Summary:**

This paper introduces a method capable of adapting a pre-trained model to a larger label space. To achieve this, it leverages the information of geometric distances between labels within the target larger space and replaces the common argmax operation with the Frechet mean. The paper also includes several theoretical analyses on sample complexity and optimal label subspaces.

Based on the proposed locus definition, the paper presents an active learning paradigm with the objective of expanding the maximum label coverage by introducing new labels. The empirical results demonstrate that the proposed method outperforms the vanilla baseline method in tasks of zero-shot classification and partially-observed label spaces.

**Strengths:**

- the usage of Frechet mean is simple and intuitive for solving partial label classification tasks
- the theoretical analyses are thorough and address several key questions in terms of sample complexity and optimal label subspace

**Weaknesses:**

- Evaluation Fairness: The empirical results from Table 1 and 2 are reported as $E[d^2(y, \hat{y})]$ in the respective metric space that Loki is optimized for. It is not too surprising that Loki outperforms the argmax/one-vs-rest baseline trained without such information. The paper lacks more fair metrics like accuracy or a held-out metric space to demonstrate the overall effectiveness of Loki.
- Stronger Baselines: Both zero-shot classification and partially-observed classification experiments only include vanilla baselines (e.g., CLIP argmax, SimCLR with one-vs-rest classifier). Considering the proposed method is pretty simple, the paper should compare against stronger baselines [1][2][3][4] in terms of performance and efficiency. Additionally, the claim that graph neural network architectures are heavyweight and challenging to scale up to extremely large graphs, where Loki performs better, needs quantitative measurements to support it (e.g., running efficiency, model size, etc.).
- Uncalibrated Uncertainty in Loki Formulation: Often, the probability of outputs from classification is uncalibrated [5], and the paper lacks an ablation study examining how the correctness of uncertainty $P_{\lambda}$ affects the final performance.
- Effectiveness of Active Learning: While the final experiment indicates that the proposed active learning paradigm successfully leads to larger loci, there is no guarantee or proof on how well the paradigm affects the final performance in classification tasks.
- Clarity: Section 4.2 is too wordy; consider emphasizing key statements and bringing several key algorithms from B.2, B.3 to the main text for better clarity.

[1] X. Wang, Y. Ye, and A. Gupta. Zero-shot recognition via semantic embeddings and knowledge graphs. In 2018 IEEE/CVF Conference on Computer Vision and Pattern Recognition (CVPR), pages 6857–6866, Los Alamitos, CA, USA, jun 2018. IEEE Computer Society.
[2] Abhinaba Roy, Deepanway Ghosal, Erik Cambria, Navonil Majumder, Rada Mihalcea, and Soujanya Poria. Improving Zero-Shot Learning Baselines with Commonsense Knowledge. Cognitive Computation, 14(6):2212–2222, November 2022.
[3] Christoph H. Lampert, Hannes Nickisch, and Stefan Harmeling. Learning to detect unseen object classes by between-class attribute transfer. In 2009 IEEE Conference on Computer Vision and Pattern Recognition, pages 951–958, 2009.
[4] Junyu Gao, Tianzhu Zhang, and Changsheng Xu. I Know the Relationships: Zero-Shot Action Recognition via Two-Stream Graph Convolutional Networks and Knowledge Graphs. Proceedings of the AAAI Conference on Artificial Intelligence, 33(01):8303–8311, July 2019.
[5] Nixon, Jeremy, Michael W. Dusenberry, Linchuan Zhang, Ghassen Jerfel, and Dustin Tran. "Measuring Calibration in Deep Learning." In CVPR workshops, vol. 2, no. 7. 2019.

**Questions:**

As mentioned in the weakness section, here are few key questions left to answer:
- How does Loki perform in terms of accuracy?
- How does Loki perform compared to stronger baselines from recent literatures?
- How does uncalibrated probability affect the final performance?
- How useful is the proposed active learning in terms of final classification performance?

**Limitations:**

The authors have adequately discussed several limitations and negative societal impact in the appendix.

---

> ### Author Rebuttal · Authors · 2023-08-09
>
> Thank you for the helpful comments, and for praising the simplicity of our method and the thoroughness of our theoretical analysis!
>
> **On evaluation fairness:**
>
> This is an excellent point – we have added experimental results that address it in two ways.
> - First, in our new calibration results, we were excited to find that if the softmax temperature is tuned, **Loki can attain a better 0-1 loss than argmax prediction!** This result is illustrated in Figures 1 and 2 of our new results.
> - Second, we provide new results for our CLIP experiments that use a normalized version of our metric that allows us to **compare across metric spaces**: $\mathbb{E}[d^2(y, \hat{y})] / \text{diam}(G)^2$. This metric allows us to compare across metric spaces by bringing all of the errors onto the same scale, and it also generalizes the 0-1 error when the complete graph is used, allowing us to compare against accuracy. See Table 1 in our new results. This new evaluation metric enables us to determine which metric spaces have geometry best ‘fit’ to our pretrained models. For example, for CIFAR-100, we observed that the WordNet metric space resulted in the lowest error and has the best geometry.
>
> **On baselines**
>
> Our work introduces a new problem setting: adapting a pretrained classifier to enable it to navigate a metric space, without any additional training. Note that **this setting is highly challenging due to the extreme paucity of resources** available for it: we do not get to train or fine-tune models, or, in most cases, observe any new data points, and sometimes cannot even access model internals such as embeddings. Existing approaches (including the suggested ones) require access to information not available in our scenario or require training a specialized model. In fact, _our motivation for developing Loki was our initial skepticism that class geometry alone was sufficient to improve pretrained models to any extent_. Fortunately, our empirical and theoretical results demonstrate that such improvements are often possible.
>
> The papers suggested do not fit our setting for the following reasons:
> - The method proposed in [1] requires class embeddings, which are sufficient but not necessary to apply Loki, and it **requires training** whereas Loki does not.
> - The zero-shot method proposed in [2] **requires training** a graph convolutional autoencoder and is limited to classes in ConceptNet.
> - The method in [3] **requires attribute metadata**, which is different from our metric space assumption.
> - [4] also **requires training**.
>
> In contrast to these existing methods, **Loki is a drop-in replacement for argmax prediction used in pretrained classifiers**. As far as we are aware, [1, 2, 3, 4] cannot operate in this setting.
>
> **However, we can evaluate alternatives to Loki** – Loki replaces the $\text{arg max}$ with the Fréchet mean. We can use a different function instead of the Fréchet mean, e.g., the Fréchet median. As described in the common response, we compared four alternatives to Loki in our new experimental results (presented in Table 2) and found that Loki was optimal for minimizing the expected squared distance.
>
> **On graph neural network architectures being heavyweight**
>
> By heavyweight, we mean that most other methods **require training, whereas ours does not**. Regarding the scaling to large graphs, we note that
>
> - A forward pass using a graph convolutional network (GCN) [5] requires time complexity $O(|E|)$, where $|E|$ is the number of edges.
> - A forward pass using Loki requires time complexity $O(K|V|)$, and if the number of observed classes, $K$, using Loki is small $(K << |V|)$, then the time complexity is $O(|V|)$.
>
> For problems where the graph is dense or when we only have access to pairwise distances in a finite metric space instead of a graph, the time complexity of using a GCN becomes $O(|E|) = O(|V|^2)$. This is a common setting that occurs, for example, when the metric space is derived from pretrained class-name embeddings. However in this setting, the time complexity of applying Loki is still $O(|V|)$ when $K$ is small. We conclude that the complexity of applying Loki is favorable in many graph settings.
>
> **On uncalibrated uncertainty**
>
> This is also an excellent point, thank you for bringing this up! Since submission, we performed calibration via temperature scaling [6]. Excitingly, we found that calibration indeed led to improvements in our CLIP experiments, and that the optimal temperature for Loki was always close to the optimal temperature for calibration. We also found that adjusting the temperature allowed us to navigate a Pareto curve between optimizing for the 0-1 loss and for the expected squared distance, and in some cases we improved both. We are excited about these new insights, which can be found in Figures 1 and 2 of our new results.
>
> **On the effectiveness of active selection**
>
> Thank you for bringing this up! We have included additional experimental results showing how our active strategy improves over a passive baseline both in terms of performance (expected squared distance) and expressivity (the size of the locus). This new result can be found in Figure 3 of our new experiments!.
>
> **On clarity**
>
> We agree! We have updated the text to reflect this feedback. Thank you!
>
> [1] https://arxiv.org/abs/1803.08035
>
> [2] https://arxiv.org/abs/2012.06236
>
> [3] https://ieeexplore.ieee.org/document/5206594
>
> [4] https://ojs.aaai.org/index.php/AAAI/article/view/4843
>
> [5] https://arxiv.org/abs/1609.02907
>
> [6] https://arxiv.org/abs/1706.04599

---

> ### Comment · Reviewer_zgY7 · 2023-08-16
>
> The authors have addressed my concerns in the rebuttal. I would like to raise my rating from 4 to 5. I strongly encourage the authors to include the additional materials including temperature tuning, additional evaluation metrics, and active selection experiments in the final version.

---

> > ### Author Response · Authors · 2023-08-18
> >
> > We will absolutely include these results in the final version, as we agree that these results have strengthened our work. Thank you for raising these points, and we appreciate your engagement with our rebuttal! Please let us know if there are any questions we can answer about our new results or anything else about the paper. Thanks!

---

### Author Rebuttal · Authors · 2023-08-09

We thank all reviewers for their comments! We particularly appreciate reviewers **praising the significance of our work** (reviewers cqEn, 4Uud, and 2m87) and **praising the simplicity of our method** (reviewers zgY7 and dZLD).

Given the novelty of our problem setting and approach, reviewers had several questions and suggestions for experimental improvements before recommending acceptance.
These helpful comments have significantly improved the quality of our submission, and we respond to each of them individually below. **We are confident that our work offers an exciting and impactful new problem that invites future study and practical application, while introducing a strong baseline--Loki**.

Before we proceed with these individual responses, we **highlight new results** added to the paper since submission, including several responding to reviewers’ questions and suggestions. These include a variety of exciting new experimental results that have significantly improved the paper:

### New results

1. **New Calibration Analysis** (reviewers zgY7 and 4Uud). We perform this analysis on CIFAR-100 with the CLIP-ResNet-50 model. Calibration uses softmax temperature scaling [1]. Our results are shown in Figures 1 and 2 of our attached document. In Figure 1, we provide reliability diagrams and plots navigating the tradeoff between the expected calibration error (ECE). From these experiments, we **obtain the following new insights**:
    - the optimal Softmax temperature for Loki is close to the optimal temperature for calibration,
    - tuning the temperature can lead to improvements in Loki’s performance, and
    - CLIP probabilities were already well-calibrated.

    In Figure 2, we scale the softmax temperature to navigate the tradeoff curve between the 0-1 loss and the expected squared distance on which Loki is evaluated. **For two of the metric spaces, we find that temperature scaling can lead to higher accuracy using Loki,** and even when this is not the case, temperature scaling can be used to trade off between the two evaluation metrics. We appreciate the suggestion from the reviewers; this analysis revealed the softmax temperature as an important hyperparameter for improving the results from Loki.

2. **Cross-Metric Space Comparison via a New Evaluation Metric** (reviewers zgY7 and 2m87). Expected squared distances cannot be directly compared across metric spaces, as they may be on different scales. Our solution is to use a form of normalization: we divide the expected squared distance by the square of the graph diameter. This brings all of the values to the 0-1 range, and since $\mathbb{E}[d^2(y, \hat{y})] / \text{diam}(G)^2$ indeed also generalizes the 0-1 error, this enables comparison between 0-1 errors and those from different metric spaces. We provide these results in Table 1, again for our CLIP experiments on CIFAR-100. This new evaluation metric enables us **to determine which metric spaces have geometry best 'fit' to our pretrained models**. For example, for CIFAR-100, we observed that the WordNet metric space resulted in the lowest error and has the best geometry.

3. **Comparison of Loki Alternatives** (reviewers zgY7, 4Uud, and dZLD). Loki is based on the Fréchet mean, which is defined as $arg min_{y \in \mathcal{Y}} \sum_{i=1}^K P_{\lambda_i|x} d^2(y, \lambda_i)$. However, this is not the only approach that can be considered. For example, the Fréchet *median*, often used in robust statistics, is defined as $arg min_{y \in \mathcal{Y}} \sum_{i=1}^K P_{\lambda_i|x} d(y, \lambda_i)$, without squaring the distances. More generally, we can define $arg min_{y \in \mathcal{Y}} \sum_{i=1}^K P_{\lambda_i|x} d^\beta(y, \lambda_i)$, and evaluate different choices of $\beta$. We conduct this experiment on ImageNet using SimCLR as our pretrained classifier with 250, 500, and 750 randomly selected classes. From this analysis, we conclude that using the **Fréchet mean is the optimal formulation for Loki**.

4. **Evaluation of Active Next-Class Selection Procedure on ImageNet** (reviewer zgY7). We had previously validated Theorem 4.8 in our submission using a toy example and showed that the size of the locus indeed increases optimally. In our new experiment, we validate that this optimal increase in locus size indeed results in improved performance over selection rounds. We do so on ImageNet (ILSVRC 2012) using SimCLR, beginning with 500 randomly sampled classes. Our passive baseline randomly selects a new class at each round, and our proposed approach actively selects the optimal next class at each round. Our results are shown in Figure 3. Over 50 selection rounds, we found that our active approach led to consistent and significant gains over the passive approach in expected squared distance. From this analysis, we have further evidence that our **active selection approach is effective for minimizing the expected squared distance over selection rounds**.

Please let us know if we can answer any questions about these new results! We are excited to engage with you all during the discussion phase.

[1] https://arxiv.org/abs/1706.04599

---

### Comment · Area_Chair_339S · 2023-08-16
**Author-Reviewer Discussion Period Closing Soon**

Thank you reviewers for your work in evaluating this submission, and thank you authors for responding to the reviewers’ questions and concerns. We are entering the final phase of the discussion period, which will run until August 21st, and some of the authors' responses have to been acknowledged by all the reviewers.

Reviewers: If you have any lingering questions or comments on the rebuttal or the responses, now is the time to express them. At the very least, please acknowledge that you have read the authors’ response to your review.

Thank you everyone for making the review process a fruitful, constructive, and civil process.

AC

---

### Decision · Program_Chairs · 2023-09-21

**Decision:**

Accept (poster)

**Comment:**

There was consensus among the reviewers that this well-written paper tackles a challenging and relevant problem (ML model extrapolation to unseen classes), that the proposed solution is compelling and practical, and that the results sufficiently justify its feasibility. Some of the reviewers raised concerns about evaluation (additional baselines, fairness of comparison, etc), but the authors were able to convincingly address most of these in their thorough and thoughtful rebuttal. While no reviewer was particularly enthusiastic about this paper, it is nevertheless a correct, well-motivated, and exceptionally well-presented piece of work that introduced an algorithm that is likely to be useful for the broad ML community, which in my opinion justifies its acceptance into the conference.